# The antisecretory peptide AF-16 may modulate tissue edema but not inflammation in experimental peritonitis induced sepsis

Annelie Barrueta Tenhunen[1,2]*, Jaap van der Heijden[2], Ivan Blokhin[2], Fabrizia Massaro[1,3], Hans Arne Hansson[4], Ricardo Feinstein[5], Anders Larsson[6], Anders Larsson[1], Jyrki Tenhunen[2]

1 Department of Surgical Sciences, Hedenstierna Laboratory, Uppsala University, Uppsala, Sweden, 2 Department of Surgical Sciences, Section of Anesthesiology and Intensive Care, Uppsala University, Uppsala, Sweden, 3 Cardiac Anesthesia and Intensive Care, Anthea Hospital, GVM Care & Research, Bari, Italy, 4 Institute of Biomedicine, University of Gothenburg, Göteborg, Sweden, 5 Department of Pathology and Wildlife Diseases, National Veterinary Institute, Uppsala, Sweden, 6 Department of Medical Sciences, Section of Clinical Chemistry, Uppsala University, Uppsala, Sweden

* annelie.barrueta@surgsci.uu.se

**Data Availability Statement:** All relevant data are within the paper and its Supporting Information files.

## Abstract

Sepsis is a life-threatening condition due to a dysregulated immunological response to infection. Apart from source control and broad-spectrum antibiotics, management is based on fluid resuscitation and vasoactive drugs. Fluid resuscitation implicates the risk of volume overload, which in turn is associated with longer stay in intensive care, prolonged use of mechanical ventilation and increased mortality. Antisecretory factor (AF), an endogenous protein, is detectable in most tissues and in plasma. The biologically active site of the protein is located in an 8-peptide sequence, contained in a synthetic 16-peptide fragment, named AF-16. The protein as well as the peptide AF-16 has multiple modulatory effects on abnormal fluid transport and edema formation/resolution as well as in a variety of inflammatory conditions. Apart from its' anti-secretory and anti-inflammatory characteristics, AF is an inhibitor of capillary leakage in intestine. It is not known whether the protein AF or the peptide AF-16 can ameliorate symptoms in sepsis. We hypothesized that AF-16 decreases the degree of hemodynamic instability, the need of fluid resuscitation, vasopressor dose and tissue edema in fecal peritonitis. To test the hypothesis, we induced peritonitis and sepsis by injecting autologous fecal solution into abdominal cavity of anesthetized pigs, and randomized (in a blind manner) the animals to intervention (AF-16, n = 8) or control (saline, n = 8) group. After the onset of hemodynamic instability (defined as mean arterial pressure < 60 mmHg maintained for > 5 minutes), intervention with AF-16 (20 mg/kg (50 mg/ml) in 0.9% saline) intravenously (only the vehicle in the control group) and a protocolized resuscitation was started. We recorded respiratory and hemodynamic parameters hourly for twenty hours or until the animal died and collected post mortem tissue samples at the end of the experiment. No differences between the groups were observed regarding hemodynamics, overall fluid balance, lung mechanics, gas exchange or histology. However, liver wet-to-dry ratio remained lower in AF-16 treated animals as compared to controls, 3.1 ± 0.4, (2.7–3.5, 95% CI, n = 8) vs 4.0 ± 0.6 (3.4–4.5, 95% CI, n = 8), $p = 0.006$, respectively. Bearing in mind the

**Funding:** This study was supported by Swedish Research Council (grant nr. X2015-99x-22731-01-4) AL, the Swedish Heart and Lung Foundation (grant nr. 20170531) AL and ALF grant of Uppsala University Hospital. The peptide AF-16 was provided by Lantmännen Medical AB, Stockholm, Sweden. The funders had no role in study design, data collection and analysis, decision to publish, or preparation of the manuscript.

**Competing interests:** HAH is a Professor Emeritus, he has a European patent, with the number EP2468292 B1 "New peptide having antisecretory activity", H.-A. Hansson, S Lange & E Jennische, date of publication 2019-12-18. The product is in development and a phase 1 study is approved by the EMA. This does not alter our adherence to PLOS ONE policies on sharing data and materials.

limited sample size, this experimental pilot study suggests that AF-16 may inhibit sepsis induced liver edema in peritonitis-sepsis.

---

## Introduction

Sepsis is defined as "life-threatening organ dysfunction caused by a dysregulated host response to infection" [1]. In septic shock profound circulatory and metabolic abnormalities contribute to an increase in mortality, with up to 40% in-hospital mortality [1–3]. Requirement of vasopressor therapy to sustain a mean arterial pressure (MAP) > 65 mmHg in combination with persistent serum lactate level > 2 mmol/L after fluid resuscitation are the clinical hallmarks of septic shock [1]. Sepsis and septic shock are common and although sepsis mortality is decreasing [4], global estimates still count for more than 30 million cases of sepsis per year with 5.3 million potential fatalities [5].

Sepsis management is based on fluid resuscitation, broad-spectrum antibiotics, source control and vasoactive drugs [6]. Administration of intravenous fluid is fundamental to maintain adequate stroke volume and perfusion pressure, but as a consequence of fluid therapy patients often present with volume overload [7]. A positive fluid balance and volume overload is associated with longer stay in intensive care, prolonged use of mechanical ventilation and increased mortality [8–11].

Antisecretory factor (AF) is a 41 kDa protein detectable in most tissues [12]. The protein is secreted to plasma and becomes activated upon exposure to e.g. bacterial toxins [13]. AF has anti-secretory and anti-inflammatory properties [12,14,15]. The biologically active site of the protein is located in a 16-peptide fragment, AF-16, with the sequence VCHSKTRSNPENNVGL [16].

AF was first described as a potent inhibitor of intestinal hypersecretion in response to Cholera toxin [17]. It has since then been discovered to have multiple modulatory effects in altered fluid transport and edema formation/resolution [18–20] as well as in a variety of inflammatory conditions [14,21,22]. We have demonstrated in a previous study that AF-16 significantly reduced the fluid accumulation in the lungs in a porcine ventilator induced lung injury model [23]. AF is constitutively expressed in macrophages and is detectable in lymphoid organs, including gut-associated lymphoid tissue, spleen and thymus. The protein also appears to modulate proliferation of T cells [15]. Upon a pro-inflammatory stimulus AF expression is increased and the protein is redistributed from the perinuclear area to the cell surface [14,15]. This results in down-regulation of the immune response. AF is also an inhibitor of Cholera toxin induced capillary leakage [24].

Sepsis consists of a dysregulation of the fine-tuned balance between the pro- and anti-inflammatory systems. It is not known if AF or AF-16 could reverse shock symptoms in sepsis. We hypothesized that the peptide AF-16 could counteract circulatory instability in a porcine model of peritonitis induced sepsis, by reducing the inflammatory response (as disclosed by histopathology and cytokines) and/or interstitial edema formation.

## Materials and methods

The study (protocol: http://dx.doi.org/10.17504/protocols.io.bdrsi56e) was approved by the Animal Ethics Committee in Uppsala (decision 5.8.18-01054/2017). The care of the animals strictly followed the National Institute of Health guide for the care and use of Laboratory animals (NIH publications No 8023, revised 1978) and all measures were taken to minimize

suffering. Each and every animal was under deep anesthesia and received continuous intravenous analgesia during the whole experiment until the time of euthanasia. After premedication and induction of anesthesia none of the animals was awake at any time point during the experiment. The study was performed at the Hedenstierna Laboratory, Uppsala University, Sweden.

## Anesthesia and instrumentation

Sixteen pigs (8 + 8) (*Sus scrofa domesticus*) (mean weight 27.3 ± 2.4 kg) of mixed Swedish, Hampshire and Yorkshire breeds of both sexes, were premedicated and sedated with Zoletil Forte (tiletamine and zolazepam) 6 mg/kg and Rompun (xylazine) 2.2 mg/kg i.m. Thereafter a peripheral intravenous catheter was introduced in an ear vein. We placed the animals in a supine position after 5–10 min and administered a bolus of fentanyl 5–10 µg/kg i.v., after which anesthesia was maintained during the whole experiment with ketamine 30 mg/kg/h, midazolam 0.1–0.4 mg/kg/h and fentanyl 4 µg/kg/h, in glucose 2.5%. Esmeron (rocuronium) 2.5 mg/kg/h was added as muscle relaxant after adequate depth of anesthesia was assured by absence of reaction to painful stimulation between the front hooves. During the first hour thirty ml/kg/h of Ringer´s acetate was infused i.v. From the second hour until induction of peritonitis Ringer´s acetate was infused at a rate of 10 ml/kg/h.

The animals were under constant observation by anesthesiologists in order to guarantee adequate depth of anesthesia and to avoid any distress related to pain or discomfort. The animals were under deep anesthesia during the whole experiment (up to 20 hours of sepsis after onset of circulatory instability). In case any suspicion of distress (shivering or asynchrony with the ventilator) we tested the adequacy of analgesia/anesthesia with a pain stimulation between the front hooves. A bolus of 100 mg ketamine i.v. was administered if the animal reacted to the stimulus.

Animals that presented with refractory shock were euthanized just prior to circulatory collapse defined as rapidly decreasing systemic arterial pressure, bradycardia and decreasing end tidal $CO_2$. The whole experiment for each and every animal, including euthanasia, was performed under full surgical anesthesia and analgesia.

After induction of anesthesia, the animals were tracheostomized, and a tube of eight mm internal diameter (Mallinckrodt Medical, Athlone, Ireland) was inserted in the trachea and connected to a ventilator (Servo I, Maquet, Solna, Sweden). Volume controlled ventilation was maintained with the following settings: tidal volume ($V_T$) 8 ml/kg, respiratory rate (RR) 25/ min, inspiratory/expiratory time (I:E) 1:2, inspired oxygen concentration ($F_IO_2$) 0.3 and positive end-expiratory pressure (PEEP) 8 cmH$_2$O; $V_T$, I:E and PEEP were maintained constant throughout the protocol. $F_IO_2$ was adjusted aiming at $PaO_2$ >10 kPa. Respiratory rate was set at 25, but adjusted to keep $PaCO_2$ <6,5 kPa.

A pulmonary artery catheter (Edwards Life-Science, Irvine CA, USA) for measurement of cardiac output (CO) and pulmonary artery pressures, and a triple lumen central venous catheter for fluid infusions were inserted via the right jugular vein. An arterial catheter for blood sampling and blood pressure measurement was inserted in the right carotid artery, and a PiCCO (pulse contour cardiac output) catheter (PV2015L20, Pulsion, Munich, Germany) was inserted in the right femoral artery for estimation of stroke volume variation (SVV) and extravascular lung water (EVLW). Blood gas analysis was executed on an ABL 3 analyzer, (Radiometer, Copenhagen, Denmark) and performed immediately after sampling. Hemoglobin and hemoglobin oxygen saturation was separately analyzed with a hemoximeter OSM 3 (Radiometer, Copenhagen, Denmark) calibrated for porcine hemoglobin.

A midline laparotomy was performed and the bladder catheterized for urinary drainage. Caecum was identified and a small incision made, feces was collected and the incision closed.

A large-bore intra-peritoneal drain was inserted, and the abdominal incision closed. Preparation and instrumentation took 84 ± 22 minutes (intervention group: 88 ± 18 minutes, control group: 81 ± 26 minutes).

## Study protocol

Preparation was followed by at least 30 min of stabilization, after which baseline measurements were performed (Fig 1). Fecal peritonitis was induced by peritoneal instillation of autologous feces (2 g/kg body weight in 200 ml warmed 5% glucose solution). The intraperitoneal drain was removed, and the abdominal wall closed. With the induction of fecal peritonitis the infusion of Ringer's Acetate was discontinued.

After peritonitis induction, animals were randomized to intervention with AF-16 (n = 8) or control group (n = 8), (block randomization: 4x4 sealed, opaque envelopes). The research team was blinded for the group allocation. Any additional fluid infusions other than anesthetics were paused at the time of peritonitis induction. Following the onset of hemodynamic instability (defined as MAP <60 mmHg for >5 min, after initial hypertension, tachycardia, high SVV and gradual decline of MAP to below 60 mmHg) the intervention group received an initial bolus of AF-16 (Batch No. 09431, KJ Ross Petersen ApS, Copenhagen, Denmark) 20 mg/kg (50 mg/ml in 0.9% saline), over duration of 10 minutes. The initial bolus dose was followed by an infusion of 40 mg/kg over 50 minutes. The control group received equal volumes of the vehicle (0.9% saline) instead. After four and eight hours the bolus dose was repeated (AF-16 or vehicle). Piperacillin/Tazobactam 2 grams in 10 ml of 0.9% saline, every 8 hours i.v. and a protocolized resuscitation were initiated following established hemodynamic instability.

Both intervention and control groups were submitted to a protocolized resuscitation aiming at a MAP > 60 mmHg. Fluid resuscitation was initiated with Ringer's Acetate 10 ml/kg/h. If signs of hypovolemia (SVV > 15% maintained for 10 min) a bolus of 150 ml Ringer's Acetate was administered. Fluid boluses were repeated until SVV was stable < 15%. When SVV

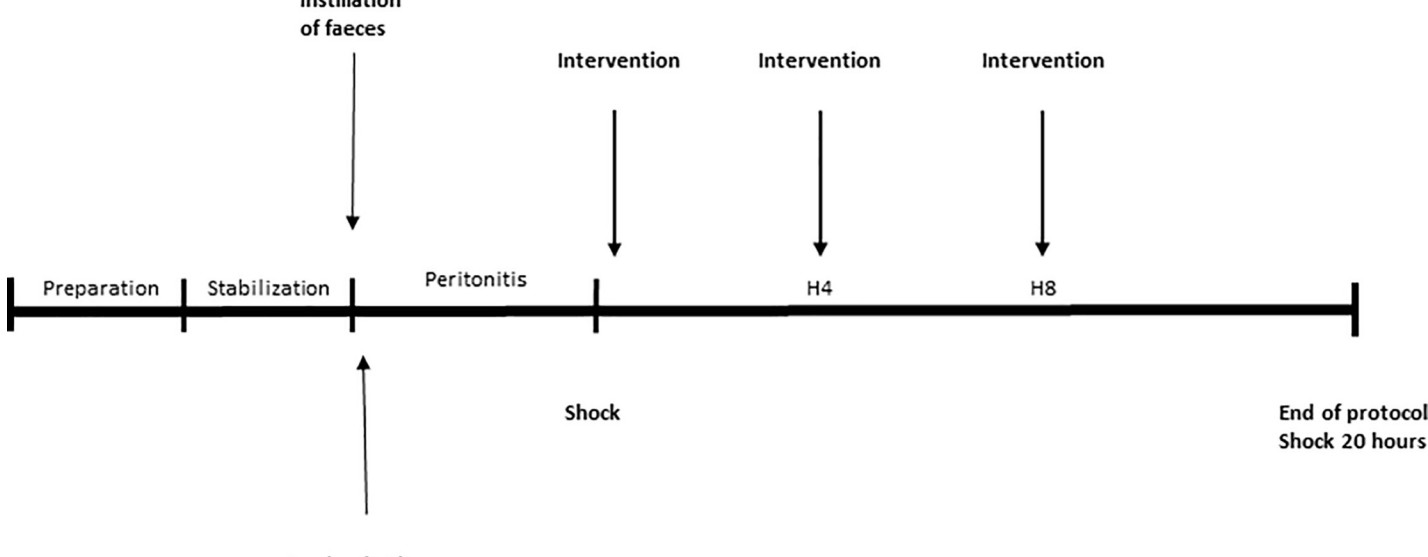

**Fig 1. Experimental time line.** After preparation and stabilization we induced peritonitis by instillation of autologous feces and the animals were randomized to intervention or control group in a blind manner. Untreated peritonitis preceded the onset of circulatory instability, when a protocolized resuscitation was initiated and intervention (or saline alone) was given at time points 0-, 4- and 8-hours. Piperacillin/Tazobactam 2 grams in 10 ml of 0.9% saline was given every 8 hours i.v. The total observation period for each animal after onset of circulatory instability was twenty hours or until death.

decreased to < 13% with MAP >60 mmHg, infusion was tapered down to 5 ml/kg/h, and if the animal was stable and SVV maintained < 13% the infusion was stopped. If signs of hypovolemia again appeared infusion was first started with 5 ml/kg/h then 10 ml/kg/h, then boluses of 150 ml were administered. In case of hypotension (MAP < 60 mmHg) without increased SVV, infusion of norepinephrine 5 ml/h (40 μg/ml) was started following a bolus of 1 ml (40 μg/ml), and increased stepwise by 5 ml/h. Glucose 30% infusion was administered, aiming at blood glucose 5–10 mmol/L, starting with 0.5 ml/kg/h. If b-glucose > 10 mmol/L an insulin infusion 1E/ml was started with 1 ml/h.

We performed blood gas analyses at baseline, after onset of shock and every hour for the following twenty hours duration of the experiment or until death. Similarly, hemodynamic parameters (systemic and pulmonary pressures, central venous pressure, CO, heart rate), respiratory parameters ($F_IO_2$, $SaO_2$, $ETCO_2$, plateau pressure, dynamic and static compliance) and hourly urine output were measured and recorded for 20 hours or until death. Every three hours, EVLW was measured and mixed venous blood gas analysis performed. Stroke volume variation (SVV) was monitored continuously in order to guide fluid administration.

The animals were euthanized with 100 mmol KCl i.v. at the end of the experiment under deep anesthesia. Thereafter the chest wall was opened. Lung tissue samples were collected from both lungs from the following regions: apical-medial, medial-medial, caudal-dorsal, caudal-medial and caudal-ventral. Samples were also taken from heart, liver, kidney, intestine and skin. The samples were immediately immersed in 10% buffered formalin. A veterinary pathologist who was blinded for the group allocation evaluated the samples histologically. Wet-to-dry ratio was measured in the above mentioned tissue samples. Samples were weighed, and dried in an oven, at 50˚C, until the weight did not differ between two measurements.

Plasma samples for analyses of IL-6 and TNF-alpha were collected at baseline, onset of circulatory instability, and then at two, four, eight hours of the observation period, and at 20 hours or immediately prior to death. IL-6 and TNF-alpha were analyzed with porcine specific sandwich ELISAs (DY686 and DY690B, R&D Systems, Minneapolis, MN, USA) according to the recommendations by the manufacturer. The total coefficient of variations (CV) for the assays were approximately 6%. All samples were analyzed at the same time. The assays were performed blinded without knowledge of clinical data.

## Statistical analysis

The Mead Resource Equation was used to determine sample size [25]. We used the Shapiro-Wilk test to test the data for normality. We compared groups with the two-tailed Student's t-test, Mann-Whitney U-test, or the Kruskal-Wallis test. Two-way repeated measures ANOVA was used to compare differences within and between the groups over time. Tukey post hoc test was applied when appropriate. Last observation carried forward was used as imputation of missing data because of early deaths. The data are expressed as mean ± SD (95% CI) or median (interquartile range) when appropriate at the baseline (following the instrumentation and stabilization, immediately prior to instillation of feces intraperitoneally), at the onset of hemodynamic instability, prior to resuscitation protocol (Sepsis 0, S0) and at the end of the experiment when the animal dies or at 20 hours of observation (End). Additionally, we present hourly recording as files (S1–S8 Appendices). The statistical analyses were conducted by SPSS v. 20.0.0 software (SPSS, Inc., Chicago, IL, USA). A $p$-value of < 0.05 was considered to be statistically significant.

## Results

Nine out of the sixteen animals survived the experiment until euthanasia (20 hours), while three and four animals died of refractory shock during the 20-hours observation period in

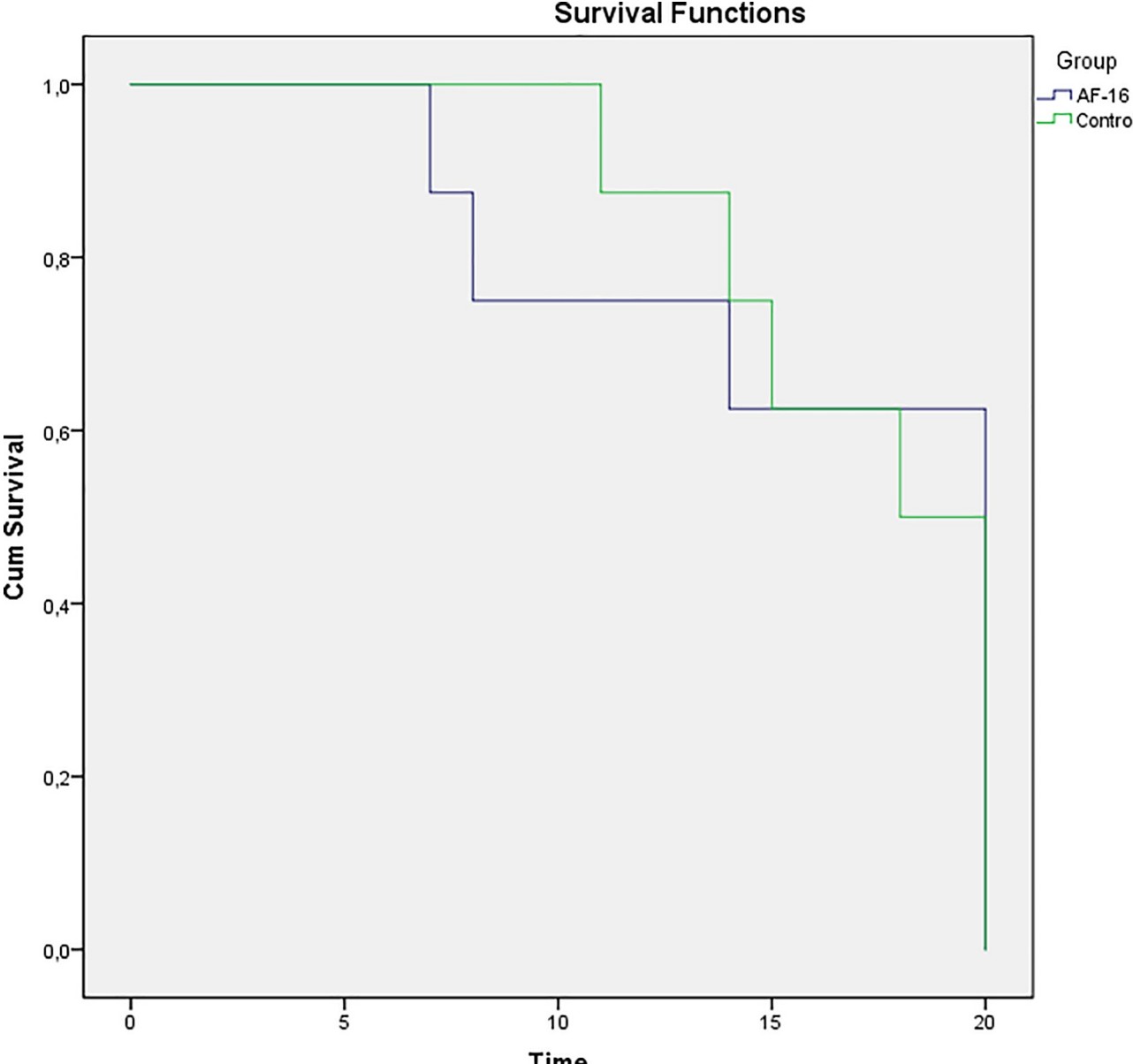

**Fig 2. Kaplan-Meyer analysis of survival.** Of a total of 8 + 8 animals, 9 survived the experiment until euthanasia (20 hours), while three and four animals died of refractory shock during the 20-hours observation period in treatment and control groups, respectively.

treatment and control groups, respectively (Fig 2). There was no statistically significant difference in survival between intervention and control groups. The results herein are presented with n = 8 per group at the baseline, at the onset of sepsis (S0) and at the End (last made observation, prior to imminent death or at 20 hours). Depiction of hourly recordings of hemodynamic and respiratory parameters and blood gas analyses are presented in the electronic supplement (S1–S8 Appendices). Comparison between the groups over time are presented herein (Two-way ANOVA).

The two groups were comparable at baseline regarding hemodynamics and respiratory parameters (Table 1). Mean time from peritonitis induction to onset of hemodynamic instability was 4.5 ± 2.2 and 4.9 ± 1.2 hours in treatment and control groups, respectively.

**Table 1. Measurements at baseline.**

| Parameter | AF-16, *n* 8 | 95% CI | Control, *n* 8 | 95% CI | *p* value |
|---|---|---|---|---|---|
| **MAP (mmHg)** | 74 ± 13 | 63–84 | 71 ± 12 | 61–81 | 0.671 |
| **HR (BPM)** | 82 ± 11 | 72–91 | 90 ± 10 | 81–99 | 0.150 |
| **CO (l/min)** | 2.4 ± 0.7 | 1.8–2.9 | 2.7 ± 0.7 | 2.1–3.3 | 0.333 |
| **MPAP (mmHg)** | 19 ± 3 | 17–21 | 18 ± 3 | 16–20 | 0.424 |
| **EVLW (ml)** | 360 ± 90 | 290–430 | 290 ± 40 | 250–320 | 0.130 |
| **SVV (%)** | 10 ± 4 | 7–13 | 8 ± 4 | 5–12 | 0.341 |
| **PaO2/F$_I$O2 (kPa)** | 60 ± 3 | 57–63 | 61 ± 5 | 57–65 | 0.630 |
| **Static compliance (ml/cmH$_2$O)** | 34 ± 6 | 28–39 | 32 ± 5 | 28–37 | 0.617 |

No difference between groups at baseline when comparing hemodynamics or respiratory parameters (two tailed t-test alt. Mann-Whitney U test), values expressed as mean ± SD (95% CI). MAP (mean arterial pressure), HR (heart rate), CO (cardiac output), MPAP (mean pulmonary arterial pressure), EVLW (extra vascular lung water), SVV (stroke volume variation) and PaO2/F$_I$O2 (the arterial oxygen tension/ inspired oxygen tension).

## Gas exchange and lung mechanics

**Gas exchange.** After established hemodynamic instability both groups presented with a decline in oxygenation (PaO$_2$/F$_I$O$_2$ ratio) from 60 ± 3 kPa at baseline, to 33 ± 14 kPa at the End of the experiment and from 61 ± 5 kPa, to 27 ± 16 kPa in AF-16 and control groups, respectively (Table 2, S1 Appendix). There was no statistically significant difference in oxygenation between the intervention and control groups (two way ANOVA $F_{(2,54)} = 0.093$, $p = 1$) as a function of time. Respiratory rate was adjusted to keep PaCO$_2$ under 6.5 kPa (Table 2).

**Lung mechanics.** Static compliance decreased from 34 ± 6 ml/cm H$_2$O at baseline in the intervention group, to 15 ± 3 ml/cm H$_2$O at the End and from 32 ± 5 ml/cm H$_2$O to 15 ± 5 ml/cm H$_2$O in the control group (Table 2, S2 Appendix), (two-way ANOVA, $F_{(21,252)} = 0.145$, $p = 1.00$). Dynamic compliance and driving pressure changed comparably in both groups during the length of the experiment (Table 2).

**Table 2. Respiratory parameters.**

| | Group | BL, *n* 8/8 | 95%CI | S0, *n* 8/8 | 95%CI | End, *n* 8/8 | 95%CI | *p* value |
|---|---|---|---|---|---|---|---|---|
| **PaO2/F$_I$O2 (kPa)** | **AF-16** | 60 ± 3 | 57–63 | 49 ± 4 | 46–52 | 33 ±14 | 22–44 | *p* = 1 |
| | **Control** | 61 ± 5 | 57–65 | 49 ± 7 | 43–54 | 27 ± 16 | 13–40 | |
| **Driving pressure (mmHg)** | **AF-16** | 7 ± 1 | 6–8 | 9 ± 2 | 8–11 | 18 ± 3 | 16–20 | *p* = 0.998 |
| | **Control** | 8 ± 3 | 5–11 | 9 ± 3 | 7–11 | 20 ± 11 | 11–29 | |
| **Static compliance (ml/cmH$_2$O)** | **AF-16** | 34 ± 6 | 28–39 | 26 ± 2 | 24–28 | 15 ± 3 | 12–17 | *p* = 1 |
| | **Control** | 32 ± 5 | 28–37 | 27 ± 6 | 22–32 | 15 ± 5 | 10–19 | |
| **Dynamic compliance (ml/cmH$_2$O)** | **AF-16** | 28 ± 4 | 25–32 | 23 ± 3 | 21–25 | 12 ± 3 | 9–14 | *p* = 1 |
| | **Control** | 26 ± 6 | 21–31 | 22 ± 5 | 18–26 | 11 ± 5 | 7–15 | |
| **Saturation (%)** | **AF-16** | 96 ± 0.3 | 96–97 | 94 ± 1 | 93–96 | 91 ± 6 | 86–96 | *p* = 0.958 |
| | **Control** | 97 ± 0.9 | 96–98 | 94 ± 1.4 | 93–96 | 85 ± 11 | 76–94 | |
| **PaCO$_2$ (kPa)** | **AF-16** | 5.5 ± 0.4 | 5.2–5.8 | 5.9 ± 0.1 | 5.8–6.0 | 4.9 ± 1.5 | 3.7–6.2 | *p* = 0.006 |
| | **Control** | 5.2 ± 0.5 | 4.8–5.6 | 5.8 ± 0.4 | 5.5–6.1 | 6.5 ± 1.2 | 5.5–7.5 | |

No statistically significant difference between groups, (two-way ANOVA) except PaCO2 (p = 0.006 –two way ANOVA) were observed. Values expressed as Mean ± SD and 95% CI. PaO2/F$_I$O2 (the arterial oxygen tension/ inspired oxygen tension), PaCO$_2$ (arterial CO$_2$ partial tension).

**Table 3. Hemodynamic parameters, blood gas analyses.**

|  | Group | BL, *n* 8/8 | 95%CI | S0, *n* 8/8 | 95%CI | End, *n* 8/8 | 95%CI | *p* value |
|---|---|---|---|---|---|---|---|---|
| **MAP (mmHg)** | AF-16 | 74 ± 13 | 63–84 | 57 ± 3 | 54–59 | 59 ± 18 | 44–74 | *p* = 0.943 |
|  | Control | 71 ± 12 | 61–81 | 57 ± 3 | 55–59 | 58 ± 15 | 46–70 |  |
| **HR (BPM)** | AF-16 | 82 ± 11 | 72–91 | 159 ± 21 | 141–176 | 117 ± 34 | 89–146 | *p* = 0.761 |
|  | Control | 90 ± 10 | 81–99 | 139 ± 32 | 112–166 | 112 ± 23 | 93–131 |  |
| **CO (l/min)** | AF-16 | 2.4 ± 0.7 | 1.8–2.9 | 2.5 ± 0.2 | 2.2–2.6 | 2.9 ±2.0 | 1.3–4.6 | *p* = 0.211 |
|  | Control | 2.7 ± 0.7 | 2.1–3.3 | 2.1 ± 0.3 | 1.9–2.4 | 2.9 ± 1.2 | 1.9–3.8 |  |
| **pH** | AF-16 | 7.44 ± 0.03 | 7.42–7.47 | 7.36 ± 0.03 | 7.33–7.39 | 7.33 ± 0.18 | 7.18–7.48 | *p* = 0.947 |
|  | Control | 7.47 ± 0.04 | 7.44–7.51 | 7.39 ± 0.04 | 7.35–7.42 | 7.24 ± 0.20 | 7.07–7.41 |  |
| **Lactate (mmol/l)** | AF-16 | 3.0 ± 1.2 | 2.0–3.9 | 2.6 ± 1.0 | 1.8–3.4 | 3.6 ± 4.0 | 0.2–6.9 | *p* = 0.644 |
|  | Control | 2.6 ± 1.0 | 1.8–3.5 | 2.2 ± 0.8 | 1.5–2.8 | 3.9 ± 4.0 | 0.5–7.3 |  |
| **Hb (g/l)** | AF-16 | 87 ± 8 | 80–93 | 119 ± 12 | 109–129 | 94 ± 10 | 86–102 | *p* = 1 |
|  | Control | 85 ± 5 | 80–89 | 122 ± 5 | 117–126 | 100 ± 6 | 95–105 |  |
| **SVV (%)** | AF-16 | 10 ± 4 | 7–13 | 21 ± 8 | 14–27 | 15 ± 8 | 9–22 | *p* = 0.249 |
|  | Control | 8 ± 4 | 5–11 | 14 ± 2 | 12–16 | 16 ± 6 | 11–22 |  |
| **EVLW (ml)** | AF-16 | 360 ± 90 | 290–430 | 400 ± 140 | 280–520 | 550 ± 370 | 240–860 | *p* = 0.622 |
|  | Control | 290 ± 40 | 250–320 | 280 ± 40 | 250–310 | 450 ± 300 | 200–700 |  |
| **ERO$_2$** | AF-16 | 0.59 ± 0.12 | 0.48–0.69 | 0.55 ± 0.07 | 0.50–0.61 | 0.49 ± 0.13 | 0.38–0.60 | *p* = 0.979 |
|  | Control | 0.49 ± 0.09 | 0.42–0.57 | 0.52 ± 0.09 | 0.44–0.59 | 0.44 ± 0.09 | 0.37–0.52 |  |
| **MPAP (mmHg)** | AF-16 | 19 ± 3 | 17–21 | 22 ± 3 | 19–25 | 29 ± 6 | 24–33 | *p* = 0.999 |
|  | Control | 18 ± 3 | 16–20 | 23 ± 4 | 20–26 | 28 ± 6 | 23–33 |  |

No statistically significant difference between groups (Two-way ANOVA). Values as mean ± SD (95% CI). MAP (mean arterial pressure), HR (heart rate), CO (cardiac output), Hb (Hemoglobin concentration), SVV (stroke volume variation), EVLW (extra vascular lung water), ERO$_2$ (the oxygen extraction ratio), MPAP (mean pulmonary arterial pressure).

## Hemodynamic parameters

**Extravascular lung water (EVLW) and Stroke volume variation (SVV).** There was no statistically significant difference in EVLW evolution between intervention and control groups as a function of time (two-way ANOVA, $F_{(7,87)}$ = 0.77, *p* = 0.614). EVLW increased from 360 ± 90 ml at baseline to 550 ± 370 ml at the end of the observation period, and from 290 ± 40 ml to 450 ± 300 ml in the intervention and control groups, respectively (Table 3). Neither was there any statistically significant difference between groups as a function of time in SVV (Table 3, S3 Appendix).

**Mean arterial blood pressure, heart rate, cardiac index, systemic vascular resistance, hemoglobin and lactate concentrations.** The onset of hemodynamic instability was defined as mean arterial pressure under 60 mmHg, both intervention and control groups presented with increases in heart rate at this stage of the experiment (Table 3, S4 Appendix, S5 Appendix). There was no statistically significant difference between groups regarding heart rate throughout the observation period (Two way ANOVA, $F_{(21,252)}$ = 0.765, *p* = 0.761). We did not observe any hemodynamic or respiratory effects during or following AF-16 infusions (S1–S6 Appendices) with the once per hour sampling rate and documentation.

Cardiac index (CI) increased from the baseline to highest measured from 82 ± 23 ml/kg/min to 151 ± 61 ml/kg/min in intervention group and from 104 ± 22 to 137 ± 23 ml/kg/min in the control group (paired samples t-test, *p* = 0.006 and *p* = 0.002, respectively). CI changed comparably over time in the two groups (two way ANOVA, $F_{(1, 28)}$ = 1.926, *p* = 0.176).

Systemic vascular resistance (SVR) declined over time in both groups (S6 Appendix), from baseline 2288 ± 630 dyn.s.cm$^{-5}$ to 1699 ± 849 dyn.s.cm$^{-5}$ (*p* = 0.031) at the End and from

**Table 4. Fluid balance.**

| | AF-16, n 8 | 95% CI | Control, n 8 | 95% CI | significance |
|---|---|---|---|---|---|
| **Fluid requirement** (ml/kg/h sepsis) | 17 ± 10 | 9–25 | 15 ± 4 | 12–18 | $p = 0.648$ |
| **Urinary output** (ml/kg/h sepsis) | 1.3 ± 1 | 0.5–2.1 | 1.5 ± 1 | 0.3–2.7 | $p = 0.729$ |
| **Fluid balance** (ml/kg/h sepsis) | 16 ± 10 | 7–25 | 14 ± 4 | 10–17 | $p = 0.834$ |
| **Norepinephrine** (μg/kg/min sepsis) | 0.62 ± 0.54 | 0.17–1.08 | 0.54 ± 0.41 | 0.20–0.88 | $p = 0.725$ |
| **Body weight gain** (kg) | 13.0 ± 3.4 | 9.5–16.5 | 14.9 ± 4.0 | 11.2–18.5 | $p = 0.386$ |

No difference between groups in fluid requirement (ml/kg/h of sepsis duration), urinary output (ml/kg/h of sepsis duration), fluid balance (ml/kg/h of sepsis duration), norepinephrine requirements (μg/kg/min of sepsis duration) or body weight gain (kg before experiment vs post mortem).

2093 ± 676 dyn.s.cm$^{-5}$ to 1326 ± 618 dyn.s.cm$^{-5}$ ($p = 0.006$) in the intervention and control groups, respectively. A one way ANOVA of SVR in intervention and control group revealed a statistically significant difference over time $p = 0.000$ and $p = 0,032$ respectively, but not between groups (two-way ANOVA: $F (21,308) = 0.751$, $p = 0.778$) as function of time. A post hoc Tukey analysis of the one way ANOVA in each group respectively revealed no statistically significant difference in SVR following administration of AF-16.

Onset of hemodynamic instability was also accompanied by an increase in hemoglobin concentration in both groups (S7 Appendix), while no statistically significant difference between groups was detected (Two way ANOVA, $F (21,252) = 0.214$, $p = 1.00$). The two groups did not differ in a statistically significant way in either lactate, pH or oxygen extraction ratio (Table 3, S8 Appendix). The seven animals that died of refractory shock, however, presented with hyperlactatemia (7.3 ± 3.2 mmol/l).

**Fluid balance.** There was no statistically significant difference between groups in fluid requirements, urinary output, fluid balance (these parameters described as ml/kg/h of sepsis duration), norepinephrine consumption (μg/kg/min of sepsis duration) or body weight gain (kg body weight after–before the experiment) (Table 4).

**Wet-to-dry ratio.** Samples from lung, skin, intestine, heart (left ventricle), kidney and liver were analyzed. Lung samples from different regions were analyzed separately and the data pooled together. Skin had the lowest water content, kidney and intestine the highest. Wet-to-dry ratio at the end of the experiment was significantly lower in liver but not in other tissues in comparison between intervention and control groups. (Table 5).

**Plasma cytokines: TNF-alpha, IL-6.** Plasma TNF-alpha concentration presented with no clear dynamics during the observation period (Table 6). Plasma IL-6 concentration increased in both intervention and control groups from the baseline to the onset of circulatory instability from 273 ± 295 pg/ml to 5851 ± 3457 pg/ml ($p = 0.003$) and from 100 pg/ml ± 0

**Table 5. Wet-to-dry ratio.**

| | AF-16 | 95% CI | Control | 95% CI | Significance |
|---|---|---|---|---|---|
| **Intestine** | 4.3 ± 1.6 | 2.9–5.7 | 4.3 ± 1.3 | 3.2–5.4 | $p = 0.990$ |
| **Heart** | 3.6 ± 0.7 | 3.0–4.2 | 3.5 ± 0.7 | 2.9–4.1 | $p = 0.699$ |
| **Kidney** | 4.5 ± 1.0 | 3.6–5.3 | 4.3 ± 0.7 | 3.8–4.9 | $p = 0.798$ |
| **Liver** | 3.1 ± 0.4 | 2.7–3.5 | 4.0 ± 0.6 | 3.4–4.5 | $p = 0.006$ |
| **Lung** | 3.4 ± 0.7 | 2.8–3.9 | 3.7 ± 0.7 | 3.0–4.3 | $p = 0.400$ |
| **Skin** | 2.0 ± 0.5 | 1.6–2.4 | 1.8 ± 0.3 | 1.5–2.0 | $p = 0.279$ |

Wet-to-dry ratio in tissue samples. Values expressed as mean ± SD (95% CI). Two tailed t-test.

**Table 6. TNF-alpha concentration.**

| Time point | Group | TNF-alpha (ng/mL) | 95% CI | N |
|---|---|---|---|---|
| BL | AF-16 | 242 ± 322 | 108–376 | 8 |
| | Control | 156 ± 147 | 22–290 | 8 |
| S 0 | AF-16 | 265 ± 213 | 132–399 | 8 |
| | Control | 275 ± 229 | 141–409 | 8 |
| S 2 | AF-16 | 241 ± 216 | 107–375 | 8 |
| | Control | 246 ± 203 | 112–379 | 8 |
| S 4 | AF-16 | 226 ± 162 | 92–360 | 8 |
| | Control | 225 ± 153 | 91–359 | 8 |
| S 8 | AF-16 | 206 ± 103 | 72–340 | 8 |
| | Control | 212 ± 179 | 79–346 | 8 |
| End | AF-16 | 180 ± 120 | 46–314 | 6 |
| | Control | 171 ± 132 | 37–305 | 8 |

Concentration of TNF-alpha in plasma (pg/ml) at different time points, values expressed as mean ± SD (95% CI). Two animals in intervention group died at time S8, these are included at time 8 as their last measurement, the other animals are represented in time point "End", representing either 20-hours time point or time of imminent death. No statistically significant difference between groups at different time points (two way ANOVA, $F_{(5, 84)} = 0{,}149$, $p = 0.98$.

to 5287 pg/ml ± 2489 ($p = 0.001$), respectively. Plasma IL-6 concentrations remained high throughout the protocol in both groups with no differences between the groups as a function of time (two way ANOVA, $F_{(5,84)} = 0.353$, $p = 0.879$ (Table 7).

**Table 7. IL-6 concentration in plasma.**

| Time point | Group | IL-6 (pg/ml) | 95% CI | N |
|---|---|---|---|---|
| BL | AF-16 | 273 ± 295 | 26–519 | 8 |
| | Control | 100 ± 0 | * | 8 |
| S 0 | AF-16 | 5851 ± 3457 | 2961–8741 | 8 |
| | Control | 5287 ± 2489 | 3206–7368 | 8 |
| S 2 | AF-16 | 7618 ± 7383 | 1446–13790 | 8 |
| | Control | 6707 ± 4510 | 2936–10478 | 8 |
| S 4 | AF-16 | 8028 ± 8589 | 847–15208 | 8 |
| | Control | 6076 ± 3535 | 3120–9031 | 8 |
| S 8 | AF-16 | 10281 ± 11498 | 669–19894 | 8 |
| | Control | 7476 ± 6320 | 2192–12760 | 8 |
| END | AF-16 | 9231 ± 12299 | 1051–19513 | 6 |
| | Control | 12677 ± 11376 | 3166–22187 | 8 |

Concentration of IL-6 in plasma (pg/ml) at different time points, values expressed as mean ± SD (95% CI). Two animals in intervention group died at time S8, these are included at time S8 as their last measurement, the other animals are represented in time point "End", representing either 20-hours time point or time of imminent death. No statistically significant difference between groups at different time points (two way ANOVA, $F_{(5, 84)} = 0.353$, $p = 0.879$.

* At baseline IL-6 concentration in the control group was <100 pg/ml in all animals (represented as 100 pg/ml in table), therefore SD and 95% CI could not be calculated. Two samples could not be obtained at the End in AF-16 group.

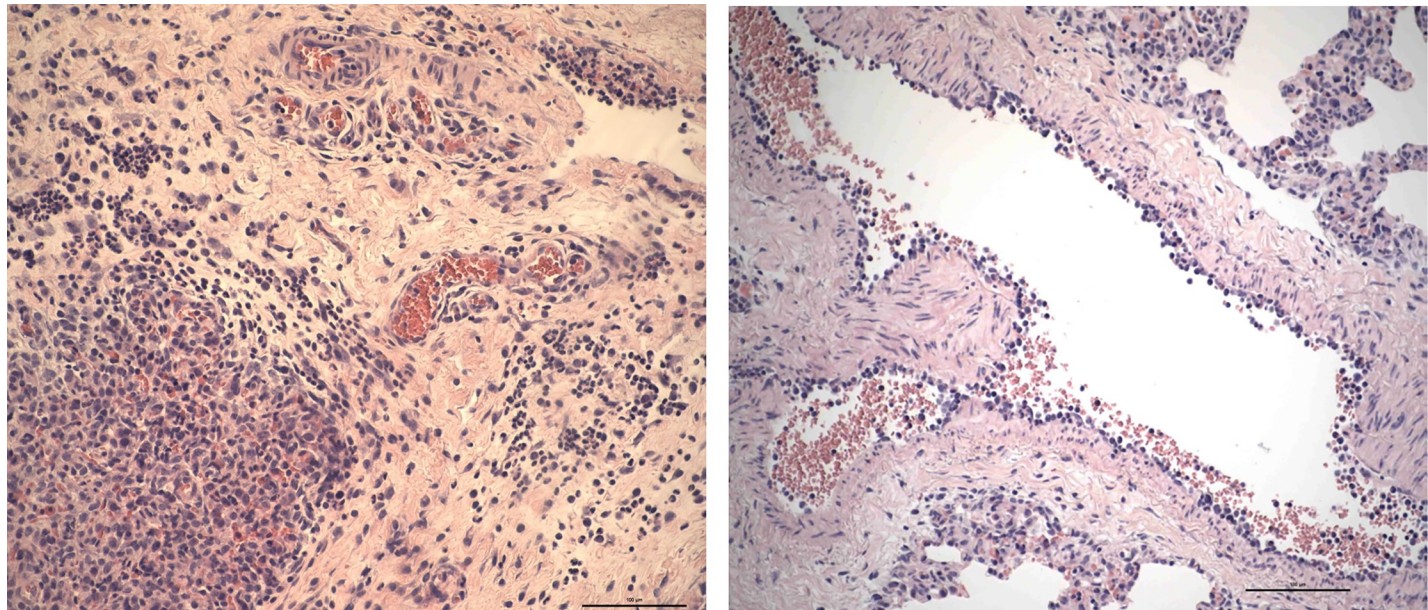

**Fig 3. Histology of lung samples, AF-16 vs control. 3A** Intense inflammatory cell reaction shows leukocytes rich in polymorphs in the alveoli (down, at left) and in the capillaries in the interlobular septum (AF-16). **3B** Bronchial vessel shows leukocytes adhering at the endothelium. It could be an early stage in the process of leukocyte migration through the vessel wall, but leukocytes seem to remain in the intima which is suggestive of endoarteriolitis, which could be predisposing for thrombosis (control).

## Histology

Abnormal lesions were found most commonly in the lungs and the intestine. The intensity of lesions was graded in a semi quantitative way, based on the numbers of inflammatory cells and the extension and distribution of the cell infiltrates/lesions. Inflammatory cell exudates in lung samples included neutrophils, monocytes and macrophages. Leucocytes were increased in the interstitium, vessels and perivascular space (Fig 3A and 3B).

Vessels often displayed prominent endothelial cells and leucocytes were found in the process of margination and migration through the vessel wall. Many lung samples showed edema, hemorrhages and recently originated micro-thrombi in small-sized vessels commonly blocking the lumen; sometimes with adjacent alveolar areas with congested septal capillaries, hemorrhages and pyknotic cells suggestive of necrosis. There was no statistically significant difference between intervention and control groups regarding inflammation or edema in lung samples (Table 8).

**Table 8. Lung histology.**

|  | Leukocytes, AF-16 | Leukocytes, Control | Atelectasis, AF-16 | Atelectasis, Control | Edema, AF-16 | Edema, Control |
|---|---|---|---|---|---|---|
| **AMR** | 1.5 (0–3) | 2.5 (1–3) | 0.5 (0–3) | 0.5 (0–4) | 2 (0–4) | 3 (0–4) |
| **MMR** | 1.5 (0–4) | 1.5 (1–4) | 0.5 (0–4) | 1 (0–3) | 1.5 (0–4) | 2 (0–4) |
| **CMR** | 1.5 (0–3) | 3 (1–4) | 4 (2–4) | 3.5 (2–4) | 2.5 (0–4) | 2.5 (1–4) |
| **CDR** | 1.5 (0–2) | 3 (1–4) | 4 (2–4) | 4 (2–4) | 2.5 (0–2) | 3 (1–4) |
| **CVR** | 1.5 (0–2) | 2.5 (0–4) | 4 (3–4) | 4 (3–4) | 2.5 (1–4) | 3.5 (1–4) |

Lung histology. Intensity of lesions: 4 very severe, 3 severe, 2 moderate, 1 mild, 0 lesions not observed. Values expressed as median (min-max). No statistically significant difference between groups, statistics presented in the following order: leucocytes, atelectasis and edema (Kruskal-Wallis $p = 0.169$, Kruskal-Wallis $p = 0.672$, Kruskal-Wallis $p = 0.751$).

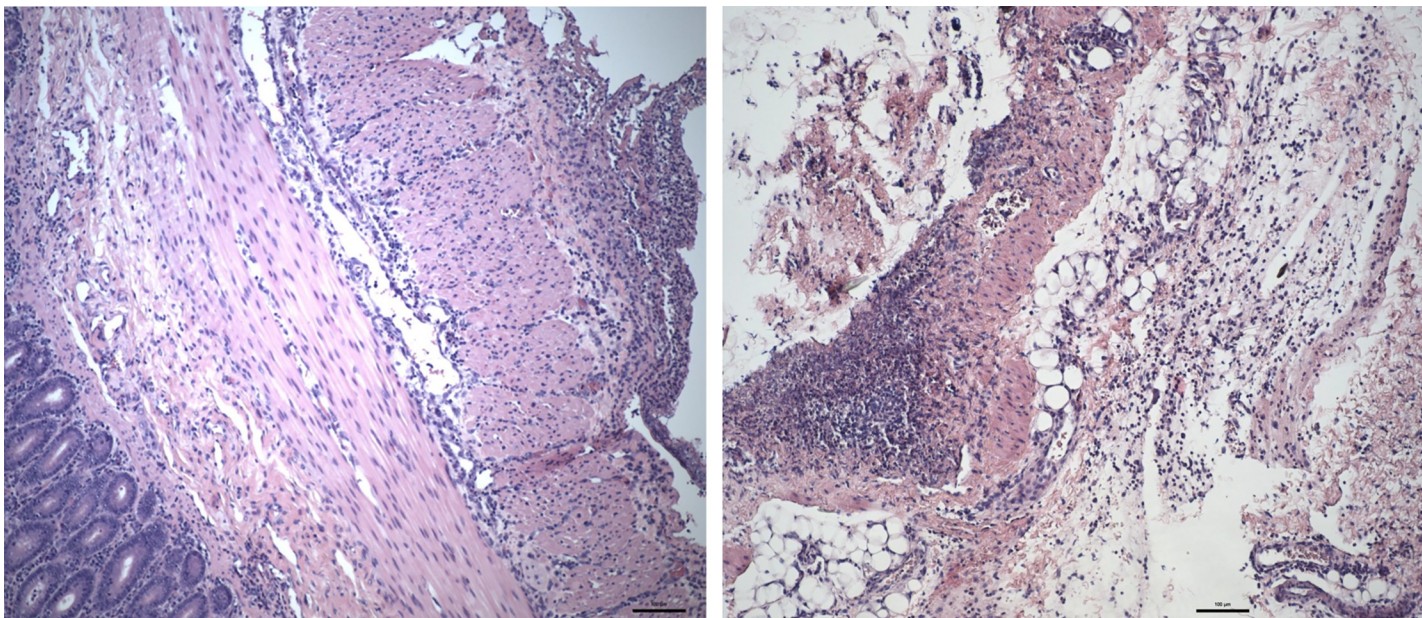

**Fig 4. Histology of intestine and mesenterium. 4A** The mucosa is down at left and the serosa is at right. The serosa shows a rich fibrinopurulent exudate, consistent with peritonitis. Leukocytes between the smooth muscle layers also are visible (AF-16). **4B** Mesenterium. Necrosis and intense inflammatory reaction in the fat and connective tissues, consistent with peritonitis (control).

The intestines showed severe acute degenerative and necrotic changes in the mucosa. Some samples of intestine showed signs of transmural inflammation (Fig 4A and 4B). There was no difference between intervention and control groups regarding signs of inflammation in samples of intestine, (Median 3, min 0, max 4 in both groups) (Kruskal-Wallis, $p = 0.321$).

There were few signs of lesions (vacuoles, inflammation) in samples of heart, liver and kidney. No lesions were detected in skin biopsies.

## Discussion

In this experimental study of peritonitis induced sepsis, intervention with the anti-secretory and anti-inflammatory peptide AF-16 did not yield any reversal of sepsis/septic shock symptoms as reflected in signs of inflammation, overall fluid balance, hemodynamics, norepinephrine consumption, gas exchange or respiratory mechanics. However, we observed in post mortem analysis of tissue wet-to-dry ratio a significantly lower wet-to-dry ratio in liver in the intervention group.

We used a model of fecal peritonitis induced sepsis, previously described by Correa et al. [26]. Prior to the main protocol we performed a pre-study of four animals, in which all 4 animals died of fulminant septic shock before finishing the planned 20-hours protocol (S11 Appendix). The animals of the pre-study presented with sepsis/septic shock after a mean duration of peritonitis of 4.25 ± 0.5 hours with mean survival 11.5 ± 4.0 hours. Although the onset of hemodynamic instability (MAP <60 mmHg for more than five minutes) was the same in the sixteen animals included in the main pilot series, sepsis was not as homogenously severe, and nine animals survived until 20 hours (euthanasia).

This model of sepsis has features in common with peritonitis induced sepsis in patients. The animals received autologous feces in the peritoneum to mimic intestinal perforation. In addition, the model is by definition a postoperative (laparotomy and instrumentation) peritonitis which is, unfortunately a relevant clinical scenario. In the present study, despite the

prompt identification of hemodynamic instability in the absence of intravenous fluid administration, followed by immediate administration of antibiotics, fluid resuscitation and infusion of noradrenalin, the mortality was substantial, 44% of the animals died before finishing the observation period of twenty hours. One of the caveats in the current study is that fecal peritonitis induced sepsis/septic shock as compared to endotoxin (LPS) models gives potentially more heterogeneous results. Thereby, the results presented herein, mostly "negative" in nature, may have been caused by the combination of large variation in the observed parameters and small sample size. As exemplified by Castellano et al. [27] LPS model may indeed be useful in search for pathophysiologic mechanisms in sepsis. We, on the other hand, chose to accept heterogeneity in order to mimic clinical sepsis. The administration of LPS, a component of the outer cell wall of gram negative bacteria, to animals or humans induces a systemic inflammatory response, with hemodynamic and biochemical abnormalities similar to those seen in sepsis and septic shock. However, according to the International Expert Consensus for Pre-Clinical Sepsis Studies [28] LPS is not an appropriate model for replicating human sepsis. More precisely, LPS challenge leads to activation of the immune system while there is no beneficial effects expected from the immune-activation. Meanwhile, infection or microbial challenge induces immunoresponse that can be both beneficial and harmful [29].

To reproduce/mimic such complex and heterogeneous conditions as human sepsis and septic shock in an experimental animal model is challenging. To evaluate a possible intervention to human sepsis in an animal model is even more challenging. "Hundreds of biological interventions have proven effective in animal models of shock and sepsis but have failed to translate to humans" [30]. The endogenous protein AF, and its active sequence AF-16, counteract edema and abnormal fluid flux [17–20,31,32]. In addition, AF protein/peptide exerts anti-inflammatory properties in a variety of conditions [15,21,22]. Neither AF nor AF-16 affect healthy tissue [32]. As for the mechanisms of action of AF and AF derived peptides, these have been studied in different models, including *in vitro* models, for instance on the modulatory effect on transport capabilities on neuronal membranes [12], and on the regulatory role of immune reactions [15]. In a rat model of cholera toxin challenge extravasation of Evans Blue was inhibited by Antisecretory factor [24]. To our knowledge however, there are no *in vitro* studies on AF-16 and endothelial cells which could be relevant in order to understand more thoroughly the mechanisms of action. Finally, considering in vitro studies as compared to animal models; the exact mechanisms of immunomodulation can be studied *in vitro*. It is reasonable to claim, however that sepsis, a condition in which microcirculation, neuroendocrine pathways, inflammatory and coagulation networks, cell death and cell protection pathways are altered, cannot be effectively studied in *in vitro* models [33,34]. In spite of the limitations, animal experimental models "remain in the critical pathway for the development of new agents for the pharmacological treatment of severe sepsis or septic shock" [29].

The definition of septic shock in humans include an increased serum lactate > 2 mmol/L despite adequate fluid resuscitation. In this study we did not observe any statistically significant hyperlactatemia in the analysis of the whole cohort or the two groups. Oxygen extraction ratio however declined significantly in both groups as a sign of either diminished oxygen demand or inefficient utilization of oxygen in the tissues. Our model of peritonitis induced sepsis/septic shock renders a heterogeneous panorama of different severity outcomes, with animals mainly represented in three different subgroups, irrespective of belonging to intervention or control group. Seven animals presented with refractory shock before the end of the observation period. These animals had high lactate and did not respond to increasing norepinephrine dosage or fluid resuscitation. The animals that survived the entire observation period of twenty hours can basically be divided into two subgroups, one group (5 animals) in which an initial hyperlactatemia resolved along with resuscitation, and another group (4 animals)

with lactate values under 2 mmol/l all along the observation period. This is in accordance with the clinical setting, where hyperlactatemia is associated with negative outcome [35–37]. Plasma IL-6 concentrations increased from the baseline to the onset of hemodynamic instability and remained high in both groups. This confirms that our peritonitis-sepsis model is a model of infection induced systemic inflammation. Administration of AF-16, however, did not modify the IL-6 response. TNF-alpha concentrations were high in both groups as compared to previously reported normal concentrations [38] already at baseline, and remained high throughout the experiment. Interestingly, Pinsky et al. have reported similar non-dynamic TNF-alpha pattern in sepsis patients [39] with corresponding TNF-alpha concentrations. AF-16 did not modify the TNF-alpha response in this study. Contrary to our expectations AF-16 did not mitigate systemic inflammation in this model of peritonitis induced sepsis. Finally, Castegren et al. have previously reported that LPS alone induces TNF-alpha release in pigs within 3 hours of LPS administration [40] potentially indicating how LPS model differs from peritonitis-sepsis model. Taken together, we wish to suggest that the current model can be considered a clinically relevant model of sepsis.

Peritonitis induced impairment of gas exchange and lung mechanics in the current study were similar to Acute Respiratory Distress Syndrome (ARDS) in humans. At the end of the protocol (after 20 hours or before the imminent death) all but three animals fulfilled oxygenation criteria for ARDS. This decrease in oxygenation was accompanied with a significant decline in both static and dynamic compliance and an increase in driving pressure with predefined tidal volumes. AF-16 did not modify the development of ARDS-like condition in this study. To add, there was no statistically significant difference in EVLW evolution between the groups during the experiment, all except three animals (one from intervention group and two control animals) did manifest an increase in EVLW, ranging from 7% to 279%.

In a previous study [23] we examined the potential effect of AF-16 on resolution of pulmonary edema in a model of ventilator induced lung injury, consisting of lung lavages and injurious ventilation. In that study a statistically significant reduction of EVLW in the intervention group was found, as an isolated finding. That finding was not reproduced in the present study, and although all animals did not respond with an increase in EVLW, leaving out the "EVLW non responders" in post-hoc analysis did not yield a different outcome. Thus, contrary to our expectation and acknowledging the limitation of low number of the animals with pronounced inter-individual variation, AF-16 did not seem to protect from increasing EVLW in peritonitis-sepsis. In models of edema and increased interstitial fluid pressure AF-16 has an early effect [18,32] on edema formation. On the other hand, in a study by Jennische et al, intranasal administration of AF-16 reduced ICP after 15 min, but no effect on inflammatory response in brain could be discerned [19]. The endogenous AF response to an inflammatory stimulus is considered to be slower. Exposure to pro-inflammatory stimulus in form of LPS or IFN-γ results in an increase in AF expression and redistribution from perinuclear area to cell surface over a time period of several days, expression peaks with severity of disease and thereafter returns to baseline. It has previously been speculated that AF plays its main role in modifying the immune reponse in the resolution phase of an inflammatory reaction, rather than at the beginning of an immunological response [12,14,15]. AF-activity is low in health and in chronic inflammatory conditions, and therefore chronic inflammation might benefit better from treatment with AF/AF-16 than acute conditions [13]. Thus, the lack of any effect of AF-16 on cytokine release in the present study may in fact not be so surprising since the observation period of sepsis/septic shock was limited to 20 hours.

In this study AF-16 was given in repeated doses, the initial dose being three times higher than in our previous ventilator induced lung injury (VILI) model [23]. Administration of AF-16 was not accompanied by any changes in hemodynamic or respiratory parameters that we

could observe. The initial administration was given at the onset of circulatory instability and minor effects/side effects might have not been noticed in the overall scenario, however at time points 4 and 8 no effects were observed during or immediately following 10-minutes infusions. This is in accordance with report by Al-Olama et al. [32] they reported no effect on systemic blood pressure associated with the administration of AF-16 in a model of tumor in rat. We cannot rule out that the intervention with AF-16 would be more effective at an even higher or continuous dose, as AF in plasma has a rapid turnover rate [12], or that an effect could have been observed in a less severe sepsis/septic shock state. Moreover, the number of animals studied was limited, and most importantly, inter-individual variation was large (SD, 95% CI) so that minor differences between the groups might not have been noticed. Neither can we rule out the possibility that AF or AF-16 could be more effective in a later stage of sepsis or septic shock with, then, potentially resolving infection. The anti-inflammatory effect of AF-16 only later follows the primary anti-secretory effect [14,15,19].

This study has limitations. No animal model reproduces the full picture of sepsis/septic shock in humans. The biological heterogeneity in sepsis patients, with differences in age, comorbidities, medications and different sources of infection adds to the complexity of the syndrome. This complexity cannot be fully represented in an animal model. In the present peritonitis/sepsis model the pigs are healthy prior to the experiment. One must also accept the possibility of interspecies variability in intestinal flora and host response to both infection and intervention. We conclude that, contrary to our hypothesis, in this pilot study in a porcine experimental model of fecal peritonitis and sepsis we could not detect any differences between intervention and control groups regarding reversal of shock symptoms, gas exchange, respiratory mechanics or overall fluid balance. However, AF-16 limited fluid accumulation, edema, in the liver. Bearing in mind the limited sample size, this experimental pilot study suggests that AF-16 may inhibit sepsis/septic shock induced liver edema in peritonitis-sepsis and therefore further studies on AF-16 in sepsis/septic shock are warranted.

## Supporting information

**S1 Appendix. $PaO_2/FIO_2$ ratio.** Decrease in $PaO_2/FIO_2$ ratio from baseline and throughout the 20 hours observation period in both groups. Reported on an hourly basis.
(TIF)

**S2 Appendix. Static compliance.** Static compliance (ml/cm $H_2O$) measured every hour of the twenty hours observation period. Decrease in compliance in both intervention and control groups.
(TIF)

**S3 Appendix. Stroke volume variation (SVV).** SVV (%) monitored continuously at the bedside and reported on an hourly basis during the 20 hours observation period in both groups.
(TIF)

**S4 Appendix. Mean arterial pressure (MAP).** MAP (mmHg) measured continuously at the bedside and recorded on an hourly basis in both groups during the 20 hours observation period.
(TIF)

**S5 Appendix. Heart rate.** Evolution of heart rate (beats per minute) at an hourly basis during the twenty hours observation period, shows an increase in heart rate in both groups.
(TIF)

**S6 Appendix. Systemic vascular resistance (SVR).** Systemic vascular resistance (SVR) calculated for intervention and control groups, respectively, on an hourly basis during the 20 hours observation period.
(TIF)

**S7 Appendix. Hemoglobin.** Hemoglobin concentration (g/l) measured every hour in both groups during the 20 hours observation period.
(TIF)

**S8 Appendix. Lactate.** Arterial blood lactate concentration (mmol/l) measured every hour in both groups during the 20 hours observation period.
(TIF)

**S9 Appendix. Fluid requirements.** Total fluid requirements during resuscitation period of maximum 20 hours, reported in ml/kg every 15 minutes.
(TIF)

**S10 Appendix. Norepinephrine consumption.** Norepinephrine consumption in μg/kg/min registered continuously and reported every 15 minutes during the observation period.
(TIF)

**S11 Appendix. Pre-study peritonitis model.** Peritonitis induced sepsis in pilot study (four pigs) to study model, pig 4* received intervention (AF-16).
(DOCX)

**S1 Data.**
(XLSX)

## Author Contributions

**Conceptualization:** Annelie Barrueta Tenhunen, Anders Larsson, Jyrki Tenhunen.

**Data curation:** Annelie Barrueta Tenhunen, Jaap van der Heijden.

**Formal analysis:** Annelie Barrueta Tenhunen, Jaap van der Heijden, Ricardo Feinstein, Anders Larsson, Jyrki Tenhunen.

**Funding acquisition:** Anders Larsson.

**Investigation:** Annelie Barrueta Tenhunen, Jaap van der Heijden, Ivan Blokhin, Fabrizia Massaro, Ricardo Feinstein, Anders Larsson, Jyrki Tenhunen.

**Methodology:** Annelie Barrueta Tenhunen, Hans Arne Hansson, Ricardo Feinstein, Anders Larsson, Anders Larsson, Jyrki Tenhunen.

**Project administration:** Annelie Barrueta Tenhunen.

**Resources:** Anders Larsson, Anders Larsson.

**Supervision:** Jyrki Tenhunen.

**Validation:** Jaap van der Heijden, Jyrki Tenhunen.

**Visualization:** Annelie Barrueta Tenhunen, Ricardo Feinstein.

**Writing – original draft:** Annelie Barrueta Tenhunen.

**Writing – review & editing:** Annelie Barrueta Tenhunen, Jaap van der Heijden, Hans Arne Hansson, Anders Larsson, Jyrki Tenhunen.

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
