## [Decision Letter · Decision Letter 0]

9 Jun 2020

PONE-D-20-08184

Does the Antisecretory Peptide AF-16 modulate fluid balance and inflammation in experimental peritonitis induced sepsis?

PLOS ONE

Dear Dr. Barrueta Tenhunen,

Thank you for submitting your manuscript to PLOS ONE. After careful consideration, we feel that it has merit but does not fully meet PLOS ONE’s publication criteria as it currently stands. Therefore, we invite you to submit a revised version of the manuscript that addresses the points raised during the review process.

We look forward to receiving your revised manuscript.

Kind regards,

Aleksandar R. Zivkovic

Academic Editor

PLOS ONE

Journal Requirements:

2. We note that you have a patent relating to material pertinent to this article. Please provide an amended statement of Competing Interests to declare this patent (with details including name and number), along with any other relevant declarations relating to employment, consultancy, patents, products in development or modified products etc. Please confirm that this does not alter your adherence to all PLOS ONE policies on sharing data and materials, as detailed online in our guide for authors http://journals.plos.org/plosone/s/competing-interests by including the following statement: "This does not alter our adherence to  PLOS ONE policies on sharing data and materials.” If there are restrictions on sharing of data and/or materials, please state these. Please note that we cannot proceed with consideration of your article until this information has been declared.

Please respond by return email and we will update the online submission form on your behalf.

3. Please ensure that you refer to Figures 4a and 4b in your text as, if accepted, production will need this reference to link the reader to the figure.

4. Please upload a copy of Figures 5 and 6, to which you refer in your text on page 13. If the figure is no longer to be included as part of the submission please remove all reference to it within the text.

Reviewers' comments:

Reviewer's Responses to Questions

**Comments to the Author**

1. Is the manuscript technically sound, and do the data support the conclusions?

Reviewer #1: Yes

Reviewer #2: Partly

Reviewer #3: Yes

Reviewer #4: Yes

2. Has the statistical analysis been performed appropriately and rigorously? 

Reviewer #1: Yes

Reviewer #2: Yes

Reviewer #3: Yes

Reviewer #4: Yes

3. Have the authors made all data underlying the findings in their manuscript fully available?

Reviewer #1: Yes

Reviewer #2: Yes

Reviewer #3: Yes

Reviewer #4: Yes

4. Is the manuscript presented in an intelligible fashion and written in standard English?

Reviewer #1: Yes

Reviewer #2: Yes

Reviewer #3: Yes

Reviewer #4: Yes

5. Review Comments to the Author

Reviewer #1: In a porcine model of peritonitis-induced sepsis, possible beneficial effects of the peptide AF-16 were investigated. No significant effects of AF-16 on hemodynamics, fluid balance, lung mechanics, gas exchange or histology were found.

Although the central result is negative, it is clinically relevant and the study was well designed and performed.

How do the authors explain the lack of effect of sepsis on lactate. In other studies of porcine peritonitis-induced sepsis hyperlactatemia was reported, what could be the difference?

Did the authors measure some inflammatory mediators (e.g. TNF-alpha, IL-6)?

In general, the data are provided only at baseline, S0 and end time points. However according to methods, most data (blood gases, hemodynamics, respiratory parameters, urine output) were measured every hour. It would be worthwhile to report sepsis development at all time points measured (at least for some parameters using graphs) and not only at those 3 time points. In porcine sepsis, hyperdynamic circulation with increased CO is usually reported, which does not seem to be the case in this study based on values provided (similar CO at all 3 time points). The complete time course points could reveal the transition from hyperdynamic to hypodynamic circulation (as suggested by heart rate values).

Furthermore, AF-16 was administered in 3 boluses, did not the authors observe some changes or increased variability of some parameters during (or short after) infusion of AF-16?

Did the authors measure the central venous pressure and (calculate) the systemic vascular resistance, which could be influenced by ANF-16?

Reviewer #2: General

Sepsis is a life threatening condition and management is based on fluid resuscitation and vasoactive drugs. It is very important research to find new tools to attenuate the risk of fluid overloads.

The researchers is trying to establish if an antisecretory Peptide AF-16 has an effect in an sepsis shock model. By the use of protocolizes resuscitation, the researchers determines degree of hemodynamic instability, the need of fluid resuscitation, vasopressor dose, tissue inflammation and edema.

Major general concerns

1. Line 195. Small number of animals/per group that survived the protocol. Total number 16 animals (both sexes). Only 9 survived the 20h protocol. “Last observation carried forward was used as imputation of missing data because of early deaths” (Statistics). This mean that the non-survived animals were included in all results (called “END”). Unclear and not discussed what this means for the interpretation. Especially if in the results terminology like “same time points” “at the end of the protocol” and “function of time” are used, it is very confusing.

2. Line 177. The Mead Resource Equation was used to determine sample size. What primary outcome and estimated SD was used to determine sample size. Was N=8 or N=4 (non-survivors excluded) the calculated sample size?

3. Title gives a question mark “?” So can we or can we not make any conclusions. The author make a conclusion in the abstract.

4. 20 hours of monitoring. However no data analyses were done about the frequency of adjustments in hemodynamics and relation to adjustment of fluid requirements. These data can indicate the degree of hemodynamic instability.

5. Table 4. Unclear if the fluid balance relates to the %weight gain

6. Table 5. The wet to dry ratio’s are odd, considering that tissue contains ± 70% water. This is = 70/30 = 2.3 wet to dry ratio. Th ratio is is expected to increase when edema is in the tissues.

Minor concerns

Abstract

7. Line 38: Methods: How many animals were finally used for the presented data (only 9 survived).

8. Line 39: Methods: “resuscitation” should be “intervention with AF-16 and a protocolized resuscitation was started…”. Include amount of AF-16 and how it was administered.

Materials and Methods

9. Line 92. Include group size

10. Line 100. Why are the animals resuscitated (substantial amounts of fluids) before the initiation of peritonitis.

11. Line 130. How long took the “preparation”

12. Figure1: Please include when/how much fluids, when antibiotics were given

13. Line 114-118. How and Are over time (20h) the hyperdamics and decisions for adjustments recorded. Were the decisions always taken by the same person (or team).

14. Line 177. What was the result of the sample size calculation. What parameters were used. On what groupsize (survived animals?) statistics were done.

15. Line 182. The use of 95% confident interval (95%CI) can be considered to indicate how confident the mean value represents the real value. SD is used to indicate the variability of the measurement.

Results

16. Start the results with line 195-198. Indicate the final N that is used in the results

17. Add number of observations (N) to all tables and figure 2.

18. Add P values of statistics in tables

19. With the use of (95%CI) in tables, the reader can interpret the data more easily.

20. Discussion

21. Line 293-312 Can the author make a conclusion if the model used is a sepsis shock model? Can the early deaths be explained by the longterm or too deep anesthesia? What is the experience of the author with this type of longterm anesthesia in healthy animals.

22. Line 317,320,326: Please add conclusion sentences

23. Line 345: Please explain the relation between the limited sample size in relation to the primary hypothesized outcome and the SD of the parameters.

24. Line 346: Not sure what you mean with “in a later stage if sepsis”. The present model is a shock model. Is it possible that it is effective in a more mild sepsis (not shock) model (e,g, ten Have etal).

Conclusion

25. Line 359: please add to last sentence “in sepsis shock”.

Reviewer #3: This paper presents the results of a negative study on the possible use of the Antisecretory Peptide AF-16 in sepsis induced by peritonitis.

The manuscript acknowledges some significant limitations of sepsis models in large animals. The lack of sustained hyperlactatemia or lactate acidosis could be due to the fact that the animals were resuscitated very early after displaying hemodynamic instability. Five minutes with MAP < 60 mmHg might be not enough to produce severe shock and tissue injury, but the mortality rate of the model (regardless of the experimental group) was likely the reason for the inability of maintaining the animal hemodynamically unstable (and therefore induce more severe shock) for longer time periods.

The results are certainly affected by this limitation and, while I am personally in favor of reporting negative results, the design of the study needs to be improved in order for the manuscript to be sounder and suitable for publication.

I suggest that the authors consider two main aspects that were not part of their study and should be considered to improve it.

1. Is there any evidence from in vitro models that AF-16 does not induce any protective effect in sepsis? The authors should attempt to include an in vitro model of sepsis and septic shock in their study, where they analyze the effect of AF-16 in cultures of the same tissues that were analyzed in their animal experiment. In addition to this, a survey of the literature should also be included in the discussion, so that a clearer picture of how exactly AF-16 may (or may not, as the data suggest) help in sepsis and septic shock would be provided. Given the legitimate doubts about the severity of their model, while considering the technical limitations due to the mortality rates, adding the perspective of in vitro studies could help clarify the mechanisms of action of AF-16. It should also be considered, though, that any data from an appropriately design in vitro study should be reconciled with the data of the in vivo model in order for the work to be cohesive.

2. A broader survey of the literature should also be carried out to discuss more thoroughly the issues related to hemodynamic instability, severity of sepsis and septic shock, impact on hyperlactatemia and lactate acidosis, and ultimately tissue/organ injury. It is possible that no tissue/organ injury (and to a greater extent dysfunction) was actually induced, and therefore the apparent ineffectiveness of AF-16 is actually due to the fact that under the experimental conditions of the study the animals did not need any treatment. It is true that the mortality rate was high, but the reasons for it may not be related to tissue injury that takes longer to develop.

Reviewer #4: This paper aims to evaluate the potential beneficial role of AF-16 in the modulation of symptoms associated to peritonitis induced sepsis. The major finding is that AF-16 treatment did not reverse

sepsis symptoms as reflected in signs of inflammation, fluid balance, hemodynamics, tissue edema, norepinephrine consumption, gas exchange or respiratory mechanics.

The paper is very clear and the research is well conducted, some minor concerns to be included in the discussion section:

Main Concerns:

1. Figure 1 and figure 2 legends are missing,

2. Authors used a model of fecal peritonitis induced sepsis, with animals receiving autologous feces in the peritoneum to mimic intestinal perforation. Given the molecular mechanism of AF-16 (Lange S, Lönnroth I. The Antisecretory Factor: Synthesis, Anatomical and Cellular Distribution, and Biological Action in Experimental and Clinical Studies. Int. Rev. Cytol. 2001;210: 39–74) the AF-16 would have provided similar results in a model of LPS injected pigs?

3. As correctly indicated, the possibility of interspecies variability in intestinal flora and host response to both infection and intervention should be taken in account. Authors should discuss about the possible advantages and disadvantages of this experimental study compared to other swine model of sepsis (i.e. LPS injected), more references should be added. The authors should cite: Castellano G et al Critical Care Volume 18, Issue 5, 2014, Article number 520

4. Furthermore, the use of both the sexes, considering the importance of sex differences in immunological activation is another element that could be considered as responsible for the poor outcome after AF-16, that is anti-inflammatory and protective

5. Beside hemodynamic changes, what about other parameters such as coagulation (Prothrombin time (PT), and activated partial thromboplastin time (PTT) and fibrinolysis, kidney function (blood creatinine and BUN) or liver function (ALT, AST)?

6. PLOS authors have the option to publish the peer review history of their article (what does this mean?). If published, this will include your full peer review and any attached files.

Reviewer #1: No

Reviewer #2: No

Reviewer #3: Yes: Federico Aletti

Reviewer #4: No

---

## [Author Response · Author response to Decision Letter 0]

28 Jul 2020

Uppsala 2020-07-23

Dear Editor and Reviewers, 

We are very grateful for the critical yet constructive feedback by the editorial office and the reviewers and highly appreciate the opportunity to send now in a revised version of our manuscript. After careful revision we have made number of changes according to suggestions by the reviewers and are happy to send in a, hopefully, improved version of the manuscript. In the following, please find our response to each question, comment and suggestion. 

Best regards, 

Annelie Barrueta Tenhunen

Thank you for the opportunity to correct our figures, we have renamed our files, and hopefully the manuscript now meets PLOS ONE’s style requirements. 

2. We note that you have a patent relating to material pertinent to this article. Please provide an amended statement of Competing Interests to declare this patent (with details including name and number), along with any other relevant declarations relating to employment, consultancy, patents, products in development or modified products etc. Please confirm that this does not alter your adherence to all PLOS ONE policies on sharing data and materials, as detailed online in our guide for authors http://journals.plos.org/plosone/s/competing-interests by including the following statement: "This does not alter our adherence to PLOS ONE policies on sharing data and materials.” If there are restrictions on sharing of data and/or materials, please state these. Please note that we cannot proceed with consideration of your article until this information has been declared.

Please respond by return email and we will update the online submission form on your behalf.

We thank for the opportunity to provide an amended statement of Competing Interests, we will also send the same information by return email.

HAH is a Professor Emeritus, he has a European patent, with the number EP2468292 B1 “New peptide having antisecretory activity”, H.-A. Hansson, S Lange & E Jennische, date of publication 2019-12-18. The product is in development and a phase 1 study is approved by the EMA. This does not alter our adherence to PLOS ONE policies on sharing data and materials. 

3. Please ensure that you refer to Figures 4a and 4b in your text as, if accepted, production will need this reference to link the reader to the figure.

Thank you for this comment, we have now added reference to Figures 4a and 4b, on p 16, line 348. 

4. Please upload a copy of Figures 5 and 6, to which you refer in your text on page 13. If the figure is no longer to be included as part of the submission please remove all reference to it within the text.

We are grateful for the opportunity to be able to correct our manuscript. We have now deleted reference to Figures 5 and 6 and correctly refer to Figures 4a and 4b.

Again, we thank for the opportunity to improve our manuscript and have added captions for the Supporting Information files at the end of our manuscript on pages 25 – 26, lines 590 – 613, and have updated in-text citations accordingly.

Review Comments to the Author:

Reviewer #1: In a porcine model of peritonitis-induced sepsis, possible beneficial effects of the peptide AF-16 were investigated. No significant effects of AF-16 on hemodynamics, fluid balance, lung mechanics, gas exchange or histology were found. Although the central result is negative, it is clinically relevant and the study was well designed and performed.

How do the authors explain the lack of effect of sepsis on lactate. In other studies of porcine peritonitis-induced sepsis hyperlactatemia was reported, what could be the difference?

We are grateful for the comment and question. In our study the heterogeneous nature of sepsis and septic shock are elucidated. As in patients, the response to infection in the animals varies. In the present study, all the animals but one, presented with elevated lactate at some time point/ time points. However, all the animals had signs of infection (high fever in addition to a known infectious insult) and scored at least 3 points in the SOFA score: we interpret this as consistent with sepsis. Arterial whole blood lactate in the current study is correlated with disease severity and mortality, with increasingly high lactate in the animals that died before the end of the observation period of 20 hours. The seven animals that died in refractory shock before finishing the observation period of twenty hours had lactate concentrations of 7.3 ± 3.2 mmol/l at the time of imminent death. Most importantly, this is in accordance with the clinical scenario, where high arterial lactate predicts/is associated with negative outcome.

Accordingly we have now added to the Results, page 12, lines 283 – 284: 

“The seven animals that died of refractory shock, however, presented with hyperlactatemia (7.3 ± 3.2 mmol/l).”

Hyperlactatemia per se in pigs in the experimental setting can be due to several other factors than sepsis severity, such as stressful handling, breed and weight difference (Hofmaier et al. Laboratory Animals. 2013). An additional aspect is that in some countries the resuscitation fluid of choice is Ringer’s Lactate, while we used Ringer’s Acetate. The infusion of lactate-containing i.v. solutions may potentially complicate the interpretation of blood lactate concentrations (Orbegozo et al. Br J Anaest. 2014).

Finally, please also see our response to comments from reviewers #3 and #4 regarding hyperlactatemia. In brief, we have added to Discussion, page 18, lines 410 – 411:

“… in the analysis of the whole cohort or the two groups.”

We have also added on page 18, lines 413 – 421:

“Our model of peritonitis induced sepsis/septic shock renders a heterogeneous panorama of different severity outcomes, with animals mainly represented in three different subgroups, irrespective of belonging to intervention or control group. Seven animals presented with refractory shock before the end of the observation period. These animals had high lactate and did not respond to increasing norepinephrine dosage or fluid resuscitation. The animals that survived the entire observation period of twenty hours can basically be divided into two subgroups, one group (5 animals) in which an initial hyperlactatemia resolved along with resuscitation, and another group (4 animals) with lactate values under 2 mmol/l all along the observation period. This is in accordance with the clinical setting, where hyperlactatemia is associated with negative outcome [35-37].”

Did the authors measure some inflammatory mediators (e.g. TNF-alpha, IL-6)?

We very much appreciate this valuable question. We have now completed our study, with the help of Prof Anders Larsson (added as co-author) with analyses of plasma TNF-alpha and IL-6 concentrations at different time points. These data with addition to the methods and relevant discussion have been incorporated to the manuscript in three different sections:

In Methods we have added at p 8, lines 185 – 191:

“Plasma samples for analyses of IL-6 and TNF-alpha were collected at baseline, onset of circulatory instability, and then at two, four, eight hours of the observation period, and at 20 hours or immediately prior to death. IL-6 and TNF-alpha were analyzed with porcine specific sandwich ELISAs (DY686 and DY690B, R&D Systems, Minneapolis, MN, USA) according to the recommendations by the manufacturer. The total coefficient of variations (CV) for the assays were approximately 6%. All samples were analyzed at the same time. The assays were performed blinded without knowledge of clinical data.”

In Results we added the following at pages 13-14, lines 304 – 322:

“Plasma cytokines: TNF-alpha, IL-6”

“Plasma TNF-alpha concentration presented with no clear dynamics during the observation period (Table 6). Plasma IL-6 concentration increased in both intervention and control groups from the baseline to the onset of circulatory instability from 273 ± 295 pg/ml to 5851 ± 3457 pg/ml (p = 0.003) and from 100 pg/ml ± 0 to 5287 pg/ml ± 2489 (p = 0.001), respectively. Plasma IL-6 concentrations remained high throughout the protocol in both groups with no differences between the groups as a function of time (two way ANOVA, F (5,84) = 0.353, p = 0.879 (Table 7).”

“Table 6. TNF-alpha concentration.

Time point Group TNF-alpha (ng/mL) 95% CI N

BL AF-16 242 ± 322 108 - 376 8

 Control 156 ± 147 22 - 290 8

S 0 AF-16 265 ± 213 132 - 399 8

 Control 275 ± 229 141 - 409 8

S 2 AF-16 241 ± 216 107 - 375 8

 Control 246 ± 203 112 - 379 8

S 4 AF-16 226 ± 162 92 - 360 8

 Control 225 ± 153 91 - 359 8

S 8 AF-16 206 ± 103 72 - 340 8

 Control 212 ± 179 79 - 346 8

End AF-16 180 ± 120 46 - 314 6

 Control 171 ± 132 37 - 305 8

Table 6. Concentration of TNF-alpha in plasma (pg/ml) at different time points, values expressed as mean ± SD (95% CI). Two animals in intervention group died at time S8, these are included at time 8 as their last measurement, the other animals are represented in time point “End”, representing either 20-hours time point or time of imminent death. No statistically significant difference between groups at different time points (two way ANOVA, F (5, 84) = 0,149, p = 0.98. Two samples could not be obtained at the End in AF-16 group. 

Table 7. IL-6 concentration in plasma

Time point Group IL-6

(pg/ml) 95% CI N

BL AF-16 273 ± 295 26 - 519 8

 Control 100 ± 0 * 8

S 0 AF-16 5851 ± 3457 2961 - 8741 8

 Control 5287 ± 2489 3206 - 7368 8

S 2 AF-16 7618 ± 7383 1446 - 13790 8

 Control 6707 ± 4510 2936 - 10478 8

S 4 AF-16 8028 ± 8589 847 - 15208 8

 Control 6076 ± 3535 3120 - 9031 8

S 8 AF-16 10281 ± 11498 669 - 19894 8

 Control 7476 ± 6320 2192 - 12760 8

END AF-16 9231 ± 12299 1051 - 19513 6

 Control 12677 ± 11376 3166 - 22187 8

Table 7. Concentration of IL-6 in plasma (pg/ml) at different time points, values expressed as mean ± SD (95% CI). Two animals in intervention group died at time S8, these are included at time S8 as their last measurement, the other animals are represented in time point “End”, representing either 20-hours time point or time of imminent death. No statistically significant difference between groups at different time points (two way ANOVA, F (5, 84) = 0.353, p = 0.879. 

*At baseline IL-6 concentration in the control group was <100 pg/ml in all animals (represented as 100 pg/ml in table), therefore SD and 95% CI could not be calculated. Two samples could not be obtained at the End in AF-16 group.” 

To the Discussion we have added to pages 18 – 19, lines 422 – 433 the following:

“Plasma IL-6 concentrations increased from the baseline to the onset of hemodynamic instability and remained high in both groups. This confirms that our peritonitis-sepsis model is a model of infection induced systemic inflammation. Administration of AF-16, however, did not modify the IL-6 response. TNF-alpha concentrations were high in both groups as compared to previously reported normal concentrations [38] already at baseline, and remained high throughout the experiment. Interestingly, Pinsky et al. have reported similar non-dynamic TNF-alpha pattern in sepsis patients [39] with corresponding TNF-alpha concentrations. AF-16 did not modify the TNF-alpha response in this study. Contrary to our expectations AF-16 did not mitigate systemic inflammation in this model of peritonitis induced sepsis. Finally, Castegren et al. have previously reported that LPS alone induces TNF-alpha release in pigs within 3 hours of LPS administration [40] potentially indicating how LPS model differs from peritonitis-sepsis model. Taken together, we wish to suggest that the current model can be considered a clinically relevant model of sepsis.”

In general, the data are provided only at baseline, S0 and end time points. However according to methods, most data (blood gases, hemodynamics, respiratory parameters, urine output) were measured every hour. It would be worthwhile to report sepsis development at all time points measured (at least for some parameters using graphs) and not only at those 3 time points. In porcine sepsis, hyperdynamic circulation with increased CO is usually reported, which does not seem to be the case in this study based on values provided (similar CO at all 3 time points). The complete time course points could reveal the transition from hyperdynamic to hypodynamic circulation (as suggested by heart rate values).

We are grateful for the comments and the suggestions. Accordingly, we have now added modified figures as supporting information files to present hourly data as recorded during the experiments. We therefore added Figures S1 Appendix – S8 Appendix as supporting data, enabling the reader to appreciate the whole time course of infection, inflammation, and sepsis. 

Since the transition from hyperdynamic to hypodynamic circulation might be “lost” in the inter-individual variation in the time course of infection and sepsis we added baseline to highest cardiac index (CI) data: Cardiac index (CI) increased from the baseline to highest measured from 82 ± 23 ml/kg/min to 151 ± 61 ml/kg/min in intervention group and from 104 ± 22 to 137 ± 23 ml/kg/min in the control group (paired samples t-test, p = 0.006 and p = 0.002, respectively). CI changed comparably over time in the two groups (two way ANOVA, F (1, 28) = 1.926, p = 0.176). 

We have accordingly added to the Results section, page 12, lines 269 – 272:

“Cardiac index (CI) increased from the baseline to highest measured from 82 ± 23 ml/kg/min to 151 ± 61 ml/kg/min in intervention group and from 104 ± 22 to 137 ± 23 ml/kg/min in the control group (paired samples t-test, p = 0.006 and p = 0.002, respectively). CI changed comparably over time in the two groups (two way ANOVA, F (1, 28) = 1.926, p = 0.176).”

Furthermore, AF-16 was administered in 3 boluses, did not the authors observe some changes or increased variability of some parameters during (or short after) infusion of AF-16?

We sincerely appreciate the possibility to be able to further clarify our observations. Administration of AF-16 was not accompanied by any changes in hemodynamic or respiratory parameters that we could observe. The initial administration was given at the onset of circulatory instability and minor effects/side effects might have not been noticed in the overall scenario, however at time points 4 and 8 no effects were observed during or immediately following the 10-minutes infusions. This is in accordance with report by Al-Olama et al. (Al-Olama et al. Acta Oncologica. 2011), they reported no effect on systemic blood pressure associated with the administration of AF-16 in a model of tumor in rat. As far as we know, AF/AF-16 has not been studied in porcine models. 

We have added to results on p 12, lines 267 – 268: 

“We did not observe any hemodynamic or respiratory effects during or following AF-16 infusions (S1 – S6 Appendix) with the once per hour sampling rate and documentation.” 

We have also added to Discussion p 20, lines 464 – 470: 

“Administration of AF-16 was not accompanied by any changes in hemodynamic or respiratory parameters that we could observe. The initial administration was given at the onset of circulatory instability and minor effects/side effects might have not been noticed in the overall scenario, however at time points 4 and 8 no effects were observed during or immediately following 10-minutes infusions. This is in accordance with report by Al-Olama et al. [32] they reported no effect on systemic blood pressure associated with the administration of AF-16 in a model of tumor in rat.”

Did the authors measure the central venous pressure and (calculate) the systemic vascular resistance, which could be influenced by AF-16?

Thank you for the question and the comment. Yes, we did measure the central venous pressure continuously and registered it on an hourly basis. In both groups SVR declined over time (paired t-test), intervention group at baseline 2288 ± 630 dyn.s.cm-5, at the end 1699 ± 849 dyn.s.cm-5 (p = 0.031), control group at baseline 2093 ± 676 dyn.s.cm-5, at the end 1326 ± 618 dyn.s.cm-5 (p = 0.006). 

A one way ANOVA of the data in intervention and control group revealed a statistically significant difference regarding time p = 0.000 and p = 0,032 in both groups respectively but not between groups (two way ANOVA). A post hoc Tukey analysis revealed no statistically significant difference in SVR following administration of AF-16. 

We have added to Results, p 12, lines 273 – 279: 

“Systemic vascular resistance (SVR) declined over time in both groups (S6 Appendix), from baseline 2288 ± 630 dyn.s.cm-5 to 1699 ± 849 dyn.s.cm-5 (p = 0.031) at the End and from 2093 ± 676 dyn.s.cm-5 to 1326 ± 618 dyn.s.cm-5 (p = 0.006) in the intervention and control groups, respectively. A one way ANOVA of SVR in intervention and control group revealed a statistically significant difference over time p = 0.000 and p = 0,032 respectively, but not between groups (two-way ANOVA: F (21,308) = 0.751, p = 0.778) as function of time. A post hoc Tukey analysis of the one way ANOVA in each group respectively revealed no statistically significant difference in SVR following administration of AF-16.”

Reviewer #2: General

Sepsis is a life threatening condition and management is based on fluid resuscitation and vasoactive drugs. It is very important research to find new tools to attenuate the risk of fluid overloads.

The researchers is trying to establish if an antisecretory Peptide AF-16 has an effect in an sepsis shock model. By the use of protocolizes resuscitation, the researchers determines degree of hemodynamic instability, the need of fluid resuscitation, vasopressor dose, tissue inflammation and edema.

Major general concerns

1. Line 195. Small number of animals/per group that survived the protocol. Total number 16 animals (both sexes). Only 9 survived the 20h protocol. “Last observation carried forward was used as imputation of missing data because of early deaths” (Statistics). This mean that the non-survived animals were included in all results (called “END”). Unclear and not discussed what this means for the interpretation. Especially if in the results terminology like “same time points” “at the end of the protocol” and “function of time” are used, it is very confusing.

We are grateful for the possibility to clarify our reasoning about imputation of data due to early deaths. Approximately 17% of the data is missing due to early deaths. This is a problem in statistical terms, since the animals in the most severe state of illness were the ones who died early and as such have missing data. If not taken into consideration, the data would erroneously only represent the less affected animals at the end. In order to compare the two groups, presuming that the animals who died would not have improved, neither spontaneously, nor with treatment, the last observation carried forward offered the best way to somehow compare the different groups, in terms of shock severity, as to hemodynamic and respiratory parameters. We chose to present the data at three time points; that is at baseline, at onset of circulatory instability and at the end, where end stands for the last observation performed before refractory shock and imminent death. This means that we included all eight animals in each group in all the analyses. 

In order to clarify our presentation of the data, as correctly pointed out by the reviewer we have changed the manuscript as follows: 

We have now added to the Methods section, pages 8-9, lines 199 – 203:

“…at the baseline (following the instrumentation and stabilization, immediately prior to instillation of feces intraperitoneally), at the onset of hemodynamic instability, prior to resuscitation protocol (Sepsis 0, S0) and at the end of the experiment when the animal dies or at 20 hours of observation (End). Additionally, we present hourly recording as supporting information files (S1-S8 Appendix).”

2. Line 177. The Mead Resource Equation was used to determine sample size. What primary outcome and estimated SD was used to determine sample size. Was N=8 or N=4 (non-survivors excluded) the calculated sample size?

Again we appreciate the question and the ability to explain how we calculated sample size. The Mead resource equation can be used when the effect of an experiment is not known a priori. According to this equation an experiment will be of an appropriate size if the error degrees of freedom is 10 < E >20. Where E = total number of animals in the experiment – the number of treatment conditions. We performed a pre-study of four animals to test the peritonitis model before starting the main series, in that small series of animals, the data of fluid requirements during the observation period after onset of circulatory instability was 15.25 ± 2.5 ml/kg/h. With a power of 0.8 and a p-value < 0.05, and a wish to detect a difference between groups of 4 ml/kg/h, we would need a number of at least 7.2 animals/group, a number that we rounded up to 8. Since the pilot series was so small, however, and a possible effect of the intervention was impossible to determine a priori, defining the main series more like a pilot study, we decided to rely on the Mead equation to explain why we chose to have eight animals in each group, that is E=16. 

The dosage of fecal load was adjusted to give a lethal model of sepsis, that is to say, degree of mortality over the length of 20 hours observation. The plan for the statistical analyses included use of imputation in form of last observation carried forward. Therefore the calculated sample size was 8 per group including the non-survivors.

3. Title gives a question mark “?” So can we or can we not make any conclusions. The author make a conclusion in the abstract.

We appreciate this comment and change the title, according to the findings of the study, after our revision and due correction after comment from reviewer #2, point 6.

P 1, line 1-3:

“The Antisecretory Peptide AF-16 may modulate tissue edema but not inflammation in experimental peritonitis induced sepsis.”

4. 20 hours of monitoring. However no data analyses were done about the frequency of adjustments in hemodynamics and relation to adjustment of fluid requirements. These data can indicate the degree of hemodynamic instability.

Thank you for this question. The adjustments in hemodynamics and relation of fluid requirements are represented in Table 4 of norepinephrine consumption and fluid administration since this was determined according to blood pressure and SVV. The protocolized resuscitation was strictly followed, and allowed for adjustments to adapt fluid and norepinephrine administration (p 7, lines 158 – 168). In order to further clarify the fluid and norepinephrine administration over the length of the experiment we have added as Supporting Information files, S9 Appendix and S10 Appendix. 

5. Table 4. Unclear if the fluid balance relates to the %weight gain

Thank you for the possibility to clarify table 4. Fluid balance refers to the difference between fluid administration and urinary output per hour of observation after onset of sepsis with circulatory instability. To clarify further the data presentation in table 4 we have now changed the data of % weight gain to absolute weight change of the animals in kilograms, as measured before experiment and post mortem.

We have accordingly changed table 4 and added to the table caption, p 13, lines 289 – 294:

“… body weight gain (kg body weight after – before the experiment) (Table 4). 

Table 4. Fluid balance

 AF-16, n 8 95% CI Control, n 8 95% CI significance

Fluid requirement

(ml/kg/h sepsis) 17 ± 10 9 – 25 15 ± 4 12 – 18 

p = 0.648

Urinary output

(ml/kg/h sepsis) 1,3 ± 1 0.5 – 2.1 1,5 ± 1 0.3 – 2.7 

p = 0.729 

Fluid balance

(ml/kg/h sepsis) 16 ± 10 7.0 – 24.5 14 ± 4 10.2 – 17.4 

p = 0.834 

Norepinephrine (µg/kg/min sepsis) 0.62 ± 0.54 0.17 – 1.08 0.54 ± 0.41 0.20 – 0.88 

p = 0.725

Body weight gain (kg) 13.0 ± 3.4 9.5 – 16.5 14.9 ± 4.0 11.2 – 18.5 p = 0.386

No difference between groups in fluid requirement (ml/kg/h of sepsis duration), urinary output (ml/kg/h of sepsis duration), fluid balance (ml/kg/h of sepsis duration), norepinephrine requirements (µg/kg/min of sepsis duration) or percent body weight gain (kg before experiment vs post mortem).”

6. Table 5. The wet to dry ratio’s are odd, considering that tissue contains ± 70% water. This is = 70/30 = 2.3 wet to dry ratio. The ratio is expected to increase when edema is in the tissues.

In a most humble, grateful tone and to our embarrassment, not least to the senior author of the manuscript (JTE): Only now, after the comment by the reviewer, we recognize an error in our calculation of the ratios. The weight of the cassette holding the tissue sample was not extracted from the whole weight. We have now recalculated the wet to dry ratios and reanalyzed the data. 

This has resulted in changes in Abstract, page 2, lines 44 – 48:

“No differences between the groups were observed regarding hemodynamics, overall fluid balance, lung mechanics, gas exchange or histology. However, liver wet-to-dry ratio remained lower in AF-16 treated animals as compared to controls, 3.1 ± 0.4, (2.7 – 3.5, 95% CI, n=8) vs 4.0 ± 0.6 (3.4 – 4.5, 95% CI, n=8), p=0.006, respectively. Bearing in mind the limited sample size, this experimental pilot study suggests that AF-16 may inhibit sepsis induced liver edema in peritonitis-sepsis.”

We have modified and corrected the Results accordingly on page 13, lines 297 – 302: 

“Samples from lung, skin, intestine, heart (left ventricle), kidney and liver were analyzed. Lung samples from different regions were analyzed separately and the data pooled together. Skin had the lowest water content, kidney and intestine the highest. Wet-to-dry ratio at the end of the experiment was significantly lower in liver but not other tissues in comparison between intervention and control groups. (Table 5).

Table 5. Wet-to-dry ratio

 AF-16 95% CI Control 95% CI Significance

Intestine 4.3 ± 1.6 2.9 – 5.7 4.3 ± 1.3 3.2 – 5.4 p = 0.990

Heart 3.6 ± 0.7 3.0 – 4.2 3.5 ± 0.7 2.9 – 4.1 p = 0.699

Kidney 4.5 ± 1.0 3.6 – 5.3 4.3 ± 0.7 3.8 – 4.9 p = 0.798

Liver 3.1 ± 0.4 2.7 – 3.5 4.0 ± 0.6 3.4 – 4.5 p = 0.006

Lung 3.4 ± 0.7 2.8 – 3.9 3.7 ± 0.7 3.0 – 4.3 p = 0.400

Skin 2.0 ± 0.5 1.6 – 2.4 1.8 ± 0.3 1.5 – 2.0 p = 0.279

Table 5. Wet-to-dry ratio. Values expressed as mean ± SD (95% CI). Two tailed t-test.”

And we have added to Discussion, pages 21, lines 484 – 489: 

“We conclude that, contrary to our hypothesis, in this pilot study in a porcine experimental model of fecal peritonitis and sepsis we could not detect any differences between intervention and control groups regarding reversal of shock symptoms, gas exchange or respiratory mechanics or overall fluid balance. However, AF-16 limited fluid accumulation, edema, in the liver. Bearing in mind the limited sample size, this experimental pilot study suggests that AF-16 may inhibit sepsis/septic shock induced liver edema in peritonitis-sepsis and therefore further studies on AF-16 in sepsis/septic shock are warranted.”

Minor concerns

Abstract

7. Line 38: Methods: How many animals were finally used for the presented data (only 9 survived).

In order to compare the two groups, we used the method of Last Observation Carried Forward as an imputation model. All eight animals in both groups, however, were alive and represented at baseline, and at onset of circulatory instability (S0). End represents the last observation before euthanasia.

We have modified and added to the Abstract the following, page 2, line 43:

“We recorded respiratory and hemodynamic parameters hourly for twenty hours or until the animal died and collected post mortem tissue samples at the end of the experiment.”

8. Line 39: Methods: “resuscitation” should be “intervention with AF-16 and a protocolized resuscitation was started…”. Include amount of AF-16 and how it was administered.

Thank you, we have made the suggested changes on page 2, lines 40 – 42: 

“…intervention with AF-16 (20 mg/kg (50 mg/ml) in 0.9% saline) intravenously (only the vehicle in the control group) and a protocolized resuscitation was started.”

Materials and Methods

9. Line 92. Include group size

Thank you, we have added on page 4, line 93: 

“(8 +8)”

10. Line 100. Why are the animals resuscitated (substantial amounts of fluids) before the initiation of peritonitis.

Thank you for the question. Our experience from these pigs that come from a farm is that the animals do not present always with an equal hydration grade. The animals do not survive anesthesia and instrumentation without fluid administration. Thus, it is necessary that all animals receive a bolus/infusion for them to endure the anesthesia, instrumentation and stabilization period. Fluid administration is paused after stabilization period prior to peritonitis induction. The effect of the period without fluid administration (during peritonitis and evolving sepsis) is seen as increasing SVV, HR and hemoglobin as in hypovolemia and capillary leakage.

11. Line 130. How long took the “preparation”

Preparation/instrumentation took 84 ± 22 minutes (intervention group: 88 ± 18 minutes, control group: 81 ± 26 minutes). 

We have accordingly added to Methods, page 6, lines 131 – 132:

“Preparation/instrumentation took 84 ± 22 minutes (intervention group: 88 ± 18 minutes, control group: 81 ± 26 minutes).”

12. Figure1: Please include when/how much fluids, when antibiotics were given

2 g of Piperacillin/Tazobactam was diluted in 10 ml of 0.9% saline. In order to clarify the amount of fluid given with the antibiotics we have added to page 7, line 155:

“… in 10 ml of 0.9% saline”

And added a caption to figure 1, accordingly, page 6, line 140 – 145: 

Fig 1. Experimental time line. After preparation and stabilization we induced peritonitis by instillation of autologous feces and the animals were randomized to intervention or control group in a blind manner. Untreated peritonitis preceded the onset of circulatory instability, when a protocolized resuscitation was initiated and intervention (or saline) was given at time points 0-, 4- and 8-hours. Piperacillin/Tazobactam 2 gram in 10 ml of 0.9% saline was given every 8 hours i.v. The total observation period after onset of circulatory instability was twenty hours.

13. Line 114-118. How and Are over time (20h) the hyperdamics and decisions for adjustments recorded. Were the decisions always taken by the same person (or team).

The whole resuscitation protocol was performed by the same team (ABT and JVH) they were both blinded for the group allocation, every change in fluid administration and norepinephrine adjustment were continuously registered at the “bedside”. In order to make this clear we have added the Supporting Information files, S9 Appendix and S10 Appendix to depict noradrenalin and fluid infusions/boluses as supporting information files.

14. Line 177. What was the result of the sample size calculation. What parameters were used. On what groupsize (survived animals?) statistics were done.

We mainly refer to the answer to question 2. The Mead resource equation can be used when the effect of an experiment is not known a priori. According to this equation an experiment will be of an appropriate size if the error degrees of freedom is 10 < E >20. Where E = total number of animals in the experiment – the number of treatment conditions. We performed a pilot study of four animals to test the peritonitis model before starting the main series, in that small series of animals, the data of fluid requirements during the observation period after onset of circulatory instability was 15.25 ± 2.5 ml/kg/h. With a power of 0.8 and a p-value < 0.05, and a wish to detect a difference between groups of 4 ml/kg/h, we would need a number of at least 7.2 animals/group, a number that we rounded up to 8. Since the pilot series was so small, however, and a potential effect could not be determined a priori, defining the main series more like a pilot study, we decided to rely on the Mead equation to explain why we chose to have eight animals in each group, that is E = 16.

Statistics were done based on n = 8+8 with the last observation carried forward imputation.

15. Line 182. The use of 95% confident interval (95%CI) can be considered to indicate how confident the mean value represents the real value. SD is used to indicate the variability of the measurement.

Thank you for this comment. In the Figures in Supporting Information files (box-plot) median, quartiles and 95%CI are depicted. We have now added to the tables 95% CI as suggested by the reviewer.

Results

16. Start the results with line 195-198. Indicate the final N that is used in the results 

We appreciate this advice and we now start the results with description of survival. We have added to the Abstract and the Methods a clarification that time point End denotes the time point of imminent death or 20-hours of observation and thus N used in the results is n, 8 + 8.

We have modified the beginning of the Results as follows, page 9, lines 208 – 215: 

“Nine out of the sixteen animals survived the experiment until euthanasia (20 hours), while three and four animals died of refractory shock during the 20-hours observation period in treatment and control groups, respectively (Fig 2). There was no statistically significant difference in survival between intervention and control groups. The results herein are presented with n=8 per group at the baseline, at the onset of sepsis (S0) and at the End (last made observation, prior to imminent death or at 20 hours). Depiction of hourly recordings of hemodynamic and respiratory parameters and blood gas analyses are presented in the electronic supplement (S1- S8 Appendix). Comparison between the groups over time are presented herein (Two-way ANOVA).”

17. Add number of observations (N) to all tables and figure 2.

Added as recommended by the reviewer.

Caption added to figure 2, page 9, lines 216 – 218:

“Fig 2. Kaplan-Meyer analysis of survival. Of a total of 8+8 animals, 9 survived the experiment until euthanasia (20 hours), while three and four animals died of refractory shock during the 20-hours observation period in treatment and control groups, respectively.”

18. Add P values of statistics in tables

Thank you. We have added p values in the tables as suggested by the reviewer.

19. With the use of (95%CI) in tables, the reader can interpret the data more easily.

We have now added 95%CI in the tables as suggested by the reviewer.

20. Discussion

21. Line 293-312 Can the author make a conclusion if the model used is a sepsis shock model? Can the early deaths be explained by the longterm or too deep anesthesia? What is the experience of the author with this type of longterm anesthesia in healthy animals.

Thank you for this comment. In this series of pigs, the animals that died before finishing the protocol all exhibited refractory shock with marked hyperlactatemia. We realize that we did not report this clearly in the first version of the manuscript. 

We have added to the results separate data on hyperlactatemia of the animals that died during the 20-hours observation period on page 12, lines 283 – 284: 

“The seven animals that died of refractory shock, however, presented with hyperlactatemia (7.3 ± 3.2 mmol/l).”

We have also added to discussion on page 18, lines 413 – 421: 

“Our model of peritonitis induced sepsis/septic shock renders a heterogeneous panorama of different severity outcomes, with animals mainly represented in three different subgroups, irrespective of belonging to intervention or control group. Seven animals presented with refractory shock before the end of the observation period. These animals had high lactate and did not respond to increasing norepinephrine dosage or fluid resuscitation. The animals that survived the entire observation period of twenty hours can basically be divided into two subgroups, one group (5 animals) in which an initial hyperlactatemia resolved along with resuscitation, and another group (4 animals) with lactate values under 2 mmol/l all along the observation period. This is in accordance with the clinical setting, where hyperlactatemia is associated with negative outcome [35-37].”

Regarding the anesthesia we are happy to further explain the experience in our team and at the Hedenstierna laboratory. Our team and the lab has long experience of anesthesia and long term experiments in pigs. The current protocol is based on an anesthetic regimen of combined ketamine/midazolam/fentanyl. Ketamine is an NMDA receptor antagonist that has a different effect profile on the cardiovascular system than for instance propofol. Ketamine causes the systemic release of catecholamines, inhibition of the vagal nerve, inhibition of norepinephrine reuptake at peripheral nerves and the myocardium, and norepinephrine release from sympathetic ganglia. Cardiovascular stimulation occurs already after small-dose ketamine infusion (Millers’ Anesthesia, eight edition, Ch 30, p 848). Modest doses of for instance Midazolam attenuate the hemodynamic effects of Ketamine. Midazolam alone causes a modest decrease in arterial blood pressure, resulting from a decrease in systemic vascular resistance. (Millers’ Anesthesia, eight edition, Ch 30, p 840). Most hemodynamic variables remain unchanged after fentanyl administration, although fentanyl has positive inotropic effects. (Millers’ Anesthesia, eight edition, Ch 31, p 879). Extensive experience at the Hedenstierna laboratory is that this regimen of anesthesia provides the best anesthesia and analgesia in the animals and that the animals very seldom show signs of distress. A ketamine based anesthesia is also the first choice in a septic patient, so the laboratory scenario more likely resembles the clinical scenario. 

In this study, unfortunately, we could not include time control animals. In earlier publications from studies performed at the same laboratory long term experiments have been performed with time control animals without adverse events (Castegren et al. PLOS ONE 2013). 

22. Line 317,320,326: Please add conclusion sentences

Thank you for the comment. We have added on page 19, line 438 – 440:

“AF-16 did not modify the development of ARDS-like condition in this study. To add, there was no statistically significant difference in EVLW evolution between the groups during the experiment.”

And on page 19, lines 447 – 449:

“Thus, contrary to our expectation and acknowledging the limitation of low number of the animals with pronounced inter-individual variation, AF-16 did not seem to protect from increasing EVLW in peritonitis-sepsis.”

23. Line 345: Please explain the relation between the limited sample size in relation to the primary hypothesized outcome and the SD of the parameters.

Thank you for the relevant comment. We have modified the sentence, now on page 20, lines 472 - 474:

“Moreover, the number of animals studied was limited, and most importantly, inter-individual variation was large (SD, 95%CI) so that minor differences between the groups might not have been noticed.”

24. Line 346: Not sure what you mean with “in a later stage if sepsis”. The present model is a shock model. Is it possible that it is effective in a more mild sepsis (not shock) model (e,g, ten Have etal).

We appreciate the opportunity to further clarify our reasoning. As discussed in the manuscript exposure to pro-inflammatory stimuli results in a redistribution of endogenous AF from the perinuclear area to the cell surface over a time period of several days, expression peaks with disease severity and thereafter returns to baseline. The effect of AF/AF-16 on edema and increased interstitial fluid pressure is seen early, whereas the anti-inflammatory response cannot be discerned in the same time frame. It has previously been speculated that AF or AF-16 plays its main role in modifying the immune reponse in the resolution phase of an inflammatory reaction, rather than at the beginning of an immunological response. The fact that AF expression peaks with severity disease and thereafter returns to baseline and since we could not report any positive effects on inflammation in our model, it is possible to speculate that an antiinflammatory effect could be found in the resolution phase of sepsis/infection, rather than in the initial extreme pro-inflammatory state. 

To explain better our idea, we have modified and added the following, p 20, lines 474 – 477: 

“Neither can we rule out the possibility that AF or AF-16 could be more effective in a later stage of sepsis or septic shock with potentially resolving infection. It has been shown previously that the anti-inflammatory effect of AF-16 only later follows the primary anti-secretory effect [14,15,19].”

Conclusion

25. Line 359: please add to last sentence “in sepsis shock”.

We have modified and added to the conclusions the following, page 21, lines 484 – 489:

“We conclude that, contrary to our hypothesis, in this pilot study in a porcine experimental model of fecal peritonitis and sepsis we could not detect any differences between intervention and control groups regarding reversal of shock symptoms, gas exchange or respiratory mechanics or overall fluid balance. However, AF-16 limited fluid accumulation, edema in the liver. Bearing in mind the limited sample size, this experimental pilot study suggests that AF-16 may inhibit sepsis/septic shock induced liver edema in peritonitis-sepsis and therefore further studies on AF-16 in sepsis/septic shock are warranted.”

Reviewer #3: This paper presents the results of a negative study on the possible use of the Antisecretory Peptide AF-16 in sepsis induced by peritonitis.

The manuscript acknowledges some significant limitations of sepsis models in large animals. The lack of sustained hyperlactatemia or lactate acidosis could be due to the fact that the animals were resuscitated very early after displaying hemodynamic instability. Five minutes with MAP < 60 mmHg might be not enough to produce severe shock and tissue injury, but the mortality rate of the model (regardless of the experimental group) was likely the reason for the inability of maintaining the animal hemodynamically unstable (and therefore induce more severe shock) for longer time periods.

The results are certainly affected by this limitation and, while I am personally in favor of reporting negative results, the design of the study needs to be improved in order for the manuscript to be sounder and suitable for publication.

I suggest that the authors consider two main aspects that were not part of their study and should be considered to improve it.

Thank you for the comments. We appreciate the critical yet constructive tone of the comments and the questions by the reviewer.

First, we recognize to our understanding the most important limitation of the study, the small sample size. Therefore, we would like to be very cautious of any strong conclusions based on this pilot study. Second, we now realize, in most humble tone, following the observations and recommendations by the reviewers, that in fact there is a significant albeit small positive signal from adequately reported tissue wet-to-dry ratios. So, we suggest that we no longer need to label the paper at hand a negative study.

To continue: Indeed, lack of marked hyperlactatemia may have been caused by early fluid resuscitation and vasopressor support. We chose to use high dosage of fecal material in order to obtain severe sepsis/septic shock with high mortality, and need of high vasopressor support and fluid resuscitation. As in clinical scenario, infection, sepsis and current treatment modalities are intertwined and potentially adding to poor outcome. “All models are wrong but some are useful”(George Box). Baring that in mind we chose knowingly a model that to the best of our understanding most potentially mimics clinical sepsis/septic shock.

In an attempt to clarify more specifically each comment: 1. Significant yet small positive signal was found when correct wet-to-dry ratios were analyzed. 2. Limited time of the hemodynamic instability prior to any treatment interventions were knowingly chosen based on the early mortality in four pre-study animals. 3. Severe untreated post-surgery peritonitis leading to sepsis and septic shock portends high mortality in patients. This is precisely the type of model we sought to have. 4. The animals were severely septic with massive fluid resuscitation and massive dosage of norepinephrine. Therefore, as by definition the animals were hemodynamically unstable throughout the observation period or until death. 5. We have now made an attempt to clarify our data analyses in that End –time point denotes either 20-hours time point during observation or the last observation at the time of imminent death. Finally, 6. We realize now, that we were unsuccessful in reporting this and have now added to the results and discussion.

Accordingly we have now added to the Results, page 12, lines 283 – 284: 

“The seven animals that died of refractory shock, however, presented with hyperlactatemia (7.3 ± 3.2 mmol/l).”

We have also added to Discussion, page 18, lines 410 – 411:

“… in the analysis of the whole cohort or the two groups.”

Finally we have added on page 18, lines 413 – 421:

“Our model of peritonitis induced sepsis/septic shock renders a heterogeneous panorama of different severity outcomes, with animals mainly represented in three different subgroups, irrespective of belonging to intervention or control group. Seven animals presented with refractory shock before the end of the observation period. These animals had high lactate and did not respond to increasing norepinephrine dosage or fluid resuscitation. The animals that survived the entire observation period of twenty hours can basically be divided into two subgroups, one group (5 animals) in which an initial hyperlactatemia resolved along with resuscitation, and another group (4 animals) with lactate values under 2 mmol/l all along the observation period. This is in accordance with the clinical setting, where hyperlactatemia is associated with negative outcome [35-37].”

1. Is there any evidence from in vitro models that AF-16 does not induce any protective effect in sepsis? The authors should attempt to include an in vitro model of sepsis and septic shock in their study, where they analyze the effect of AF-16 in cultures of the same tissues that were analyzed in their animal experiment. In addition to this, a survey of the literature should also be included in the discussion, so that a clearer picture of how exactly AF-16 may (or may not, as the data suggest) help in sepsis and septic shock would be provided. Given the legitimate doubts about the severity of their model, while considering the technical limitations due to the mortality rates, adding the perspective of in vitro studies could help clarify the mechanisms of action of AF-16. It should also be considered, though, that any data from an appropriately design in vitro study should be reconciled with the data of the in vivo model in order for the work to be cohesive.

Again, thank you for the relevant comments. First, we strongly oppose the notion that the model was not a severe model of peritonitis induced sepsis. As discussed earlier, the individual animals that died before the 20-hours observation period died of septic shock or with septic shock with massive fluid resuscitation and massive dosage of vasopressor infusion. We hope to have successfully clarified this in the revised manuscript.

AF-16 is extensively studied in several inflammatory conditions and in different conditions of altered fluid transport, hypersecretion and capillary leakage. Studies have been made with in vitro models, animal experiments and clinical trials. As an example, Davidson & Hickey measured the expression of antisecretory factor in macrophages after stimulation with LPS and IFN γ. After the exposure the surface expression of antisecretory factor was significantly upregulated. 

We have now added to discussion on pages 17 – 18, lines 391 – 408: 

“To reproduce/mimic such complex and heterogeneous conditions as human sepsis and septic shock in an experimental animal model is challenging. To evaluate a possible intervention to human sepsis in an animal model is even more challenging. “Hundreds of biological interventions have proven effective in animal models of shock and sepsis but have failed to translate to humans” [30]. The endogenous protein AF, and its active sequence AF-16, counteract edema and abnormal fluid flux [17-20,31,32]. In addition, AF protein/peptide exerts anti-inflammatory properties in a variety of conditions [15,21,22]. Neither AF nor AF-16 affect healthy tissue [32]. As for the mechanisms of action of AF and AF derived peptides, these have been studied in different models, including in vitro models, for instance on the modulatory effect on transport capabilities on neuronal membranes [12], and on the regulatory role of immune reactions [15]. In a rat model of cholera toxin challenge extravasation of Evans Blue was inhibited by Antisecretory factor [24]. To our knowledge however, there are no in vitro studies on AF-16 and endothelial cells which could be relevant in order to understand more thoroughly the mechanisms of action. Finally, considering in vitro studies as compared to animal models; the exact mechanisms of immunomodulation can be studied in vitro. It is reasonable to claim, however that sepsis, a condition in which microcirculation, neuroendocrine pathways, inflammatory and coagulation networks, cell death and cell protection pathways are altered, cannot be effectively studied in in vitro models [33,34]. In spite of the limitations, animal experimental models “… remain in the critical pathway for the development of new agents for the pharmacological treatment of severe sepsis or septic shock” [29].”

To our knowledge AF or AF-16 has never been tested in a sepsis model. Sepsis and septic shock are conditions where dysregulated inflammation, altered fluid balance, uncontrolled vasodilation and capillary leakage are intertwined. Therefore, based on the pre-existing literature, we hypothesized that AF-16 potentially has beneficial effect on the pathophysiology in sepsis. The exact mechanisms of AF-16 are not entirely clarified. At the same time, the exact pathophysiology of sepsis and septic shock is not fully elucidated either. Consequently, we strongly feel that the only possible way to test our hypothesis (that AF-16 decreases the degree of hemodynamic instability, the need of fluid resuscitation, vasopressor dose and tissue edema and inflammation in fecal peritonitis) was to perform an animal study. Furthermore, as pathophysiology in sepsis and septic shock is complicated/modified with potential iatrogenesis by currently recommended treatments the only reasonable setting is large animal trial.

Now, that we in fact have found a positive signal in our study a natural continuation will be to try to elucidate further putative mechanisms in in-vitro models, as reviewer #3 kindly suggests. For the present report however, it is currently beyond the scope of our team’s abilities to complete the report with in vitro studies. Furthermore, since the septic condition is extremely complex and inflammation is both detrimental and protective, even if an effect is discerned at a cellular level, the possibility of an improved outcome at the whole organism/animal/patient level is not certain. 

2. A broader survey of the literature should also be carried out to discuss more thoroughly the issues related to hemodynamic instability, severity of sepsis and septic shock, impact on hyperlactatemia and lactate acidosis, and ultimately tissue/organ injury. It is possible that no tissue/organ injury (and to a greater extent dysfunction) was actually induced, and therefore the apparent ineffectiveness of AF-16 is actually due to the fact that under the experimental conditions of the study the animals did not need any treatment. It is true that the mortality rate was high, but the reasons for it may not be related to tissue injury that takes longer to develop.

Thank you for this comment, we are happy for the opportunity to clarify the diversity of hemodynamic response and lactatemia in this study. 1. The cut off limit of mean arterial blood pressure <60 mmHg > five minutes was not arbitrarily chosen. The cut-off value should be seen in the setting of a progressive decline in arterial blood pressure preceded by a phase of several hours of untreated peritonitis. The period of untreated peritonitis was characterized by a significant increase in body temperature, increase in heart rate and initial hypertension followed by successive decline in arterial blood pressure (which is typical in clinical sepsis, N.B.). When compensatory mechanisms were exhausted the arterial blood pressure started to decrease and from our earlier experience of the pre-study (4 animals) MAP < 60 mmHg is the limit where the compensatory mechanisms are exhausted and prompt intervention with vasopressor therapy and fluid resuscitation is needed to avoid circulatory collapse. 2. All animals in the experiment also presented with a successive increase in hemoglobin concentration as peritonitis and sepsis/septic shock developed, as an indirect measure of capillary leakage, compatible with the picture of sepsis/septic shock. 3. Stroke volume variation (SVV) increased significantly in both groups from baseline to onset of circulatory instability as a sign of absolute or relative hypovolemia. Although the individual animals presented with different severity of the septic condition, all were in need of vasopressor and fluid therapy. 4. All the animals that died before finishing the observation period presented with refractory shock; that is, in spite of increasing doses of vasopressor and fluid administration the animals presented with circulatory collapse with extreme hyperlactatemia (S1 Appendix – S8 Appendix). 

Even minor increases of lactate concentrations in sepsis are associated with higher mortality rates, and the degree of increase in lactate concentrations is directly related to the severity of the shock state and to mortality rates (Vincent et al. Critical Care. 2016). Regardless of the mechanism of production, hyperlactatemia and especially the persistence of hyperlactatemia remains an excellent prognostic marker in critical illness. 

As to our study the sixteen animals of the main series showed a heterogeneous lactate profile. The animals that presented with refractory shock and died before finishing the protocol presented with extreme hyperlactatemia, see above. In the animals that survived the entire observation period two different patterns could be discerned, one group (5 animals) in which an initial hyperlactatemia resolved along with resuscitation, and another group (4 animals) with lactate values under 2 mmol/l all along the observation period. The definition of sepsis do not include hyperlactatemia, but is defined as “life-threatening organ dysfunction caused by a dysregulated host response to infection… organ dysfunction can be represented by an increase in the Sequential Organ Failure Assessment (SOFA) score of 2 points or more” (Singer et al. Sepsis 3. 2016). In our study the fact that all animals needed norepinephrine at the onset of circulatory instability, gives a SOFA score of 3, a value that together with infection (high fever and a known infectious insult) is enough to define the animals as in a state of sepsis. On the other hand, the fact that the animals that did not show high lactate levels survived the entire protocol, and the ones with increasing and high lactate died before the protocol was finished, is in accordance with the clinical understanding of critical illness, where the degree of increase in lactate concentrations is directly related to the severity of the shock state and to mortality rates. 

In order to improve the manuscript according to the valuable feedback from reviewer #3 we have made the following changes to the manuscript and added to the manuscript, as described above in the results section on page 18, lines 410 – 411 and to the discussion, on page 18, lines 413 – 421. 

Reviewer #4: This paper aims to evaluate the potential beneficial role of AF-16 in the modulation of symptoms associated to peritonitis induced sepsis. The major finding is that AF-16 treatment did not reverse

sepsis symptoms as reflected in signs of inflammation, fluid balance, hemodynamics, tissue edema, norepinephrine consumption, gas exchange or respiratory mechanics.

The paper is very clear and the research is well conducted, some minor concerns to be included in the discussion section:

Main Concerns:

1. Figure 1 and figure 2 legends are missing,

Thank you for this observation. We have now added legends to figure 1 and figure 2 on page 6, lines 140 – 145 and on page 9, lines 216 – 218, respectively:

“Fig 1. Experimental time line. After preparation and stabilization we induced peritonitis by instillation of autologous feces and the animals were randomized to intervention or control group in a blind manner. Untreated peritonitis preceded the onset of circulatory instability, when a protocolized resuscitation was initiated and intervention (or saline) was given at timepoints 0-, 4- and 8-hours. Piperacillin/Tazobactam 2 gram in 10 ml of 0.9% saline was given every 8 hours i.v. The total observation period after onset of circulatory instability was twenty hours.”

“Fig 2. Kaplan-Meyer analysis of survival. Of a total of 8+8 animals, 9 survived the experiment until euthanasia (20 hours), while three and four animals died of refractory shock during the 20-hours observation period in treatment and control groups, respectively.”

2. Authors used a model of fecal peritonitis induced sepsis, with animals receiving autologous feces in the peritoneum to mimic intestinal perforation. Given the molecular mechanism of AF-16 (Lange S, Lönnroth I. The Antisecretory Factor: Synthesis, Anatomical and Cellular Distribution, and Biological Action in Experimental and Clinical Studies. Int. Rev. Cytol. 2001;210: 39–74) the AF-16 would have provided similar results in a model of LPS injected pigs?

Thank you for this comment. The anti-inflammatory effect of AF-16 in intestinal inflammation is well established in both experimental models and in clinical trials. Antisecretory Factor mRNA is expressed in lymphocytes along the entire gastrointestinal tract (Lange and Lönnroth. Int Rev Cytol. 2001). But AF/AF-16 also have an effect on capillary leakage in the gut, as well as antisecretory effects. The intravenous administration of LPS triggers a strong inflammatory response, thus it is reasonable to hypothesize that AF-16 may attenuate the inflammatory cascades in an LPS model. On the other hand, now we have completed the present study with TNF-alpha and IL-6 analyses and found that AF-16 did not modify cytokine response. 

The ethical dilemma of performing animal studies requires a thorough discussion and consideration of the most meaningful (and limited) use of animals in order to be able to make conclusions about a possible benefit for patients. While LPS infusion is a well-established model of sepsis, we opted for a less used, but nevertheless theoretically clinically more relevant model. In our model, the animals at the baseline already have suffered a surgical trauma (laparotomy) in addition to cannulation etc. Therefore, the animals at baseline already show some signs of pathophysiological stress, as would be the case in post-surgical intestinal perforation. Our rational for the current model was that any potential positive effect of AF-16 in an LPS model would be difficult to extrapolate to patients, whereas any positive results in this model could provide us or others a stronger basis to continue with another pre-clinical study and ultimately a clinical trial.

Please see the following section, regarding further clarification in the manuscript.

3. As correctly indicated, the possibility of interspecies variability in intestinal flora and host response to both infection and intervention should be taken in account. Authors should discuss about the possible advantages and disadvantages of this experimental study compared to other swine model of sepsis (i.e. LPS injected), more references should be added. The authors should cite: Castellano G et al Critical Care Volume 18, Issue 5, 2014, Article number 520

Thank you for this comment, we appreciate the possibility to further evolve the reasoning behind our sepsis model. The article mentioned by reviewer #4 is very interesting and we most sincerely appreciate the recommendation. It is reasonable to assume that LPS model produces less heterogeneous results in search for pathophysiologic mechanisms. In the mentioned study in female pigs of a weight of 58.4 ± 14.7 kg, anesthesia was maintained with propofol (5-8 mg/kg/h) and (fentanyl 10 µg/kg/h), sepsis was induced with an LPS infusion. In the study design we have major differences from our model that renders a comparison of the two protocols difficult to make. The pigs are of female gender, the weight of the animals is twice the size as in our model, anesthesia is maintained with high doses of propofol (5-8 mg/kg/h) and fentanyl, while our model apart from fentanyl included ketamine and midazolam, and the sepsis induction is quite different with intravenous LPS infusion vs live bacteria (with LPS) in a fecal solution instilled in the peritoneal cavity. As already discussed in response to reviewer #2 the hemodynamic effects of ketamine and propofol are different, with propofol giving rise to a dose-dependent decrease in arterial blood pressure through a decrease in cardiac output and systemic vascular resistance, while ketamine has a direct cardiodepressant, negative inotropic effect, combined with an indirect stimulatory effect secondary to activation of the sympathetic system. Where catecholamine stores are intact the impact on hemodynamics of ketamine is much less pronounced than with a propofol based anesthesia. 

We have now added to the Discussion on page 17, lines 378 – 390:

“One of the caveats in the current study is that fecal peritonitis induced sepsis/septic shock as compared to endotoxin (LPS) models gives potentially more heterogeneous results. Thereby, the results presented herein, mostly “negative” in nature, may have been caused by the combination of large variation in the observed parameters and small sample size. As exemplified by Castellano et al. [27] LPS model may indeed be useful in search for pathophysiologic mechanisms in sepsis. We, on the other hand, chose to accept heterogeneity in order to mimic clinical sepsis. The administration of LPS, a component of the outer cell wall of gram negative bacteria, to animals or humans induces a systemic inflammatory response, with hemodynamic and biochemical abnormalities similar to those seen in sepsis and septic shock. However, according to the International Expert Consensus for Pre-Clinical Sepsis Studies [28] LPS is not an appropriate model for replicating human sepsis. More precisely, LPS challenge leads to activation of the immune system while there is no beneficial effects expected from the immune-activation. Meanwhile, infection or microbial challenge induces immunoresponse that can be both beneficial and harmful [29].”

4. Furthermore, the use of both the sexes, considering the importance of sex differences in immunological activation is another element that could be considered as responsible for the poor outcome after AF-16 that is anti-inflammatory and protective

Very true. The predominant gender in preclinical animal models is male although in the clinics almost 50% of septic patients are female (Martin et al. N Engl J Med. 2003). We based our experiment on previously published peritonitis model where both genders were used (Correa et al. Crit Care. 2013). However, in the current study all but one animal supplied by the farmer were males. 

5. Beside hemodynamic changes, what about other parameters such as coagulation (Prothrombin time (PT), and activated partial thromboplastin time (PTT) and fibrinolysis, kidney function (blood creatinine and BUN) or liver function (ALT, AST)?

This series of animals was our first series using this protocol of peritonitis induced sepsis. Even though we performed a pre-study of four animals to test the model, we still considered the main series, to some extent, to be a pilot study and therefore limited our focus to hemodynamics, fluid balance, ABGs and histology and now, after recommendation from the reviewer #1, cytokines. After this study we have repeated the same protocol with another intervention, with more extensive laboratory analyses of for instance kidney function, blood and organ cultures. This material is not yet published, but positive blood cultures and/or organ cultures were observed in all the included 16 animals (predominantly E.coli). 

If considered necessary for the manuscript to be accepted for publication we can supply with additional analyses of such as AST, ALT, Crea, Urea and Fibrin, fibrinogen in plasma samples that we have saved.

---

## [Editor Report · Decision Letter 1]

3 Aug 2020

The antisecretory peptide AF-16 may modulate tissue edema but not inflammation in experimental peritonitis induced sepsis.

PONE-D-20-08184R1

Dear Dr. Barrueta Tenhunen,

We’re pleased to inform you that your manuscript has been judged scientifically suitable for publication and will be formally accepted for publication once it meets all outstanding technical requirements.

Kind regards,

Aleksandar R. Zivkovic

Academic Editor

PLOS ONE
---

## [Editor Report · Acceptance letter]

7 Aug 2020

PONE-D-20-08184R1 

The antisecretory peptide AF-16 may modulate tissue edema but not inflammation in experimental peritonitis induced sepsis. 

Dear Dr. Barrueta Tenhunen:

I'm pleased to inform you that your manuscript has been deemed suitable for publication in PLOS ONE. Congratulations! Your manuscript is now with our production department. 

Kind regards, 

on behalf of

Dr. Aleksandar R. Zivkovic 

Academic Editor

PLOS ONE